# Global high-resolution fire-sourced PM$_{2.5}$ concentrations for 2000-2023

Yonghang Hu[1#], Chenguang Tian[1#], Xu Yue[1*], Yadong Lei[2], Yang Cao[3], Rongbin Xu[4], Yuming Guo[4]

[1] Jiangsu Key Laboratory of Atmospheric Environment Monitoring and Pollution Control, Collaborative Innovation Center of Atmospheric Environment and Equipment Technology, School of Environmental Science and Engineering, Nanjing University of Information Science & Technology (NUIST), Nanjing 210044, China
[2] State Key Laboratory of Severe Weather & Key Laboratory of Atmospheric Chemistry of CMA, Chinese Academy of Meteorological Sciences, Beijing 100081, China
[3] Jiangsu Nanjing Environmental Monitoring Center, Nanjing 210041, China
[4] Climate, Air Quality Research Unit, School of Public Health and Preventive Medicine, Monash University, Melbourne, Australia

*Correspondence to*: Xu Yue (yuexu@nuist.edu.cn)

[#] These authors contributed equally to this work.

**Abstract.** Fires are a significant disturbance in Earth's systems. Smoke aerosols emitted from fires can cause environmental degradation and climatic perturbations, leading to exacerbated air pollution and posing hazards to public health. However, research on the climatic and health impacts of fire emissions is severely limited by the scarcity of air pollution data directly attributed to these emissions. Here, we develop a global daily fire-sourced PM$_{2.5}$ concentration ([PM$_{2.5}$]) dataset at a spatial resolution of 0.25$^{\text{o}}$ for the period 2000-2023, using the GEOS-Chem chemical transport model driven with two fire emission inventories, the Global Fire Emissions Database version 4.1 with small fires (GFED4.1s) and the Quick Fire Emission Dataset version 2.5r1 (QFED2.5). Simulated all-source [PM$_{2.5}$] are bias-corrected using a machine learning algorithm, which incorporates ground observations from over 9000 monitoring sites worldwide. Then the simulated ratios between fire- and all-source [PM$_{2.5}$] at individual grids are applied to derive fire-sourced [PM$_{2.5}$]. Globally, the average fire-sourced [PM$_{2.5}$] is estimated to be 2.04 μg m$^{-3}$ with GFED4.1s and 3.96 μg m$^{-3}$ with QFED2.5. Both datasets show consistent spatial distributions with regional hotspots in central Africa and widespread decreasing trends over most areas. While the mean levels of fire-sourced [PM$_{2.5}$] are much larger at low latitudes, fire episodes at the boreal regions can cause comparable PM$_{2.5}$ levels as in the tropics. This dataset serves as a valuable tool

for exploring the impacts of fire-related air pollutants on climate, ecosystems, and public health, enabling accurate assessments and supports for decision-making in environmental management and policy.

## 1 Introduction

Atmospheric particulate matter, typically $PM_{2.5}$ with aerodynamic diameter less than 2.5 μm, poses significant impacts on air quality, climate system, and public health (Gu et al., 2021;Salana et al., 2024;Xie et al., 2024). These fine particles originate from a variety of sources, among which the biomass burning from both natural wildfires and anthropogenic activities makes substantial contributions (Burke et al., 2023;Atuyambe et al., 2024;Connolly et al., 2024). Exposure to elevated fire $PM_{2.5}$ has been observed to increase the mortality rates for various diseases, particularly cardiovascular and respiratory ailments (Chen et al., 2021). Additionally, the increased temperatures associated with wildfire emissions promote the likelihood of adverse health effects (Xu et al., 2020). Furthermore, the heterogenous distribution of fire $PM_{2.5}$ concentrations ($[PM_{2.5}]$) poses much larger population exposure for low-income countries (Xu et al., 2023). Over the past a few decades, the frequency and magnitude of wildfire occurrences have escalated due to climate change and extreme weather events (Ward et al., 2012;Zhu et al., 2021;Melia et al., 2022;Hu et al., 2024b). Hence, a comprehensive examination of the trends and influencing factors related to fire $PM_{2.5}$ is important for the development of effective environmental protection and health policies.

Currently, two principal approaches are used for estimating fire $[PM_{2.5}]$ on the large scale (Yue et al., 2024). The first method derives the changes of air pollutants before and after specific fire events using observational records. For example, Roberts and Wooster (2021) used Copernicus Atmosphere Monitoring Service, a system integrating remote sensing and ground-based observations, to estimate that fire air pollutants result in approximately 750,000 deaths annually worldwide with the highest mortality rates observed in Asia and Africa. Burke et al. (2023) utilized both surface and spaceborne $PM_{2.5}$ measurements (from 2000 to 2022) and found that wildfire smoke has stabilized or even reversed the decreasing trends of $[PM_{2.5}]$ in most U.S. states since 2016. Notably, these fire-induced increases are

expected to remain unregulated as the climate continues to warm. In addition to station and satellite observations, numerical modelling is also a valuable tool to assess [PM$_{2.5}$] from fire emissions. For instance, Chen et al. (2021) employed the GEOS-Chem model to estimate daily [PM$_{2.5}$] attributable to wildfires and revealed that short-term exposure to fire PM$_{2.5}$ increases mortality risks, especially the all-cause, cardiovascular, and respiratory deaths. Zhang et al. (2023) utilized an advanced model to assess daily [PM$_{2.5}$] originating from both fire and non-fire sources across various regions of the USA. Their findings showed that fire smokes accounted for over 25% of the daily PM$_{2.5}$ levels recorded in the Air Quality System of Environmental Protection Agency (EPA) from 2007 to 2018, deteriorating the U.S. air quality particularly along the Pacific coast and in the southeast.

However, there are considerable discrepancies in modelled concentrations of wildfire pollutants due to variations in physicochemical processes, model resolutions, and meteorological forcings (Wolke et al., 2012;Markakis et al., 2015). Moreover, differences in fire emission inventories can significantly influence the assessment of fire air pollutants. For example, Desservettaz et al. (2022) used the chemical transport model (CTM) GEOS-Chem and revealed that the Global Fire Emissions Database version 4 with small fires (GFED4s) outperformed other fire inventories in Australia. On the global scale, Pan et al. (2020) used six fire emission inventories to drive GEOS-Chem model and found that simulations using the Quick Fire Emission Dataset (QFED) version 2.4 yielded the closest estimate of aerosol optical depth compared to both site-level and satellite-based observations during fire seasons. These studies revealed certain discrepancies among fire inventories and suggested a need of the comparison among these inventories to improve the accuracy of predicted fire air pollutants.

Due to inherent limitations in CTMs and inventories, there has been a growing utilization of machine learning algorithms in recent years. These algorithms have proven effective in reducing biases in model simulations, particularly in regional analysis and prediction of wildfire pollutant concentrations. For instance, Wei et al. (2019) developed a space-time random forest (RF) algorithm that integrated satellite data, ground observations, and model outputs to estimate daily PM$_{2.5}$ and black carbon concentrations at a one-kilometer resolution across the U.S. during 2000-2020. On the global scale, Xu et al. (2023) used the RF algorithm to bias-correct the GEOS-Chem output, so as to assess global daily [PM$_{2.5}$] generated from wildfires during 2001-2019. They improved the determination

coefficient of simulated PM$_{2.5}$ from 0.22 of the original GEOS-Chem model to 0.75 with the RF adjustment. Their analyses showed that approximately 2.18 billion people experienced at least one day of severe pollution per year due to fire-emitted PM$_{2.5}$, with an average exposure of 9.9 days per person each year. However, due to the high computational cost, most CTM simulations have been performed at the regional scale or driven with a single fire inventory, limiting the ability of machine learning methods to accurately constrain fire-related air pollutants on global and long-term scales.

In this study, the GEOS-Chem model was employed to create two global datasets of fire [PM$_{2.5}$] corresponding to the GFED4.1s and QFED2.5 emission inventories. These model datasets were refined using the eXtreme Gradient Boosting (XGBoost) machine learning approach in combination with abundant *in situ* measurements from thousands of monitoring stations. Subsequently, two sets of daily fire PM$_{2.5}$ data were generated with temporal coverage from 2000 to 2023 and a fine spatial resolution of 0.25°×0.25°. We aim to systematically compare [PM$_{2.5}$] across different regions, specifically focusing on the variations in PM$_{2.5}$ levels attributable to fire emissions.

## 2 Data and methods

### 2.1 Observations of surface PM$_{2.5}$ concentrations

We collected site-level measurements of [PM$_{2.5}$] from a total of ~9000 monitoring stations in the world. The site number varied year by year with the maximum of 9541 in the year 2022. The data at 1822 sites in China for 2014-2023 were obtained from the China National Environmental Monitoring Center (CNEMC, http://www.cnemc.cn). For the earlier years (2000-2013), we interpolated the data-fusion product of Tracking Air Pollution (TAP, http://tapdata.org.cn) (Geng et al., 2021;Xiao et al., 2021) to 1822 ground stations to align with the CNEMC data. This TAP dataset has shown a good consistency with observed Chinese PM$_{2.5}$ levels for 2015-2022 (Fig. S1). In the United States, PM$_{2.5}$ observations at 1198 sites for 2000-2023 were obtained from the Environmental Protection Agency (EPA, https://www.epa.gov). European PM$_{2.5}$ data at 1687 sites in 2013-2023 were obtained from the European Environment Agency (EEA, https://www.eea.europa.eu/en), with data in pre-2013 coming from the European Monitoring and Evaluation Programme (EMEP, https://emep.int). PM$_{2.5}$ data for

other countries at 4995 sites were downloaded from OpenAQ (https://openaq.org) and the World's Air Pollution: Real-time Air Quality Index (https://waqi.info), where the Air Quality Index (AQI) was converted to $PM_{2.5}$ following a standardized methodology (Benchrif et al., 2021). All $PM_{2.5}$ data, both daily and hourly, have undergone rigorous quality checks with outliers removed to ensure accuracy. Daily values were calculated from the hourly data, and sites with fewer than 10-day data in a year were excluded from the analysis. Finally, a total of 2560645 records were compiled for the model training and validation.

## 2.2 Auxiliary data

The auxiliary data utilized in this study are detailed in Table S1. Climatic data were downloaded from the ECMWF Reanalysis v5 (ERA5, https://www.ecmwf.int/en/forecasts/dataset/ecmwf-reanalysis-v5) with a spatial resolution of 0.25°×0.25° at the hourly interval. Relative humidity was calculated using surface pressure, 2-m temperature, and 2-meter dewpoint, based on the Clausius-Clapeyron equation (Pechony and Shindell, 2009). These meteorological data were aggregated to daily time scale to be consistent with $PM_{2.5}$ measurements. Global land cover data from 2000 to 2023 were obtained from MODIS Land Cover (https://modis-land.gsfc.nasa.gov/landcover.html), which classifies 17 vegetation types according to the International Geosphere-Biosphere Programme (IGBP).

## 2.3 GEOS-Chem model simulation

We used the GEOS-Chem model (version 12.0.0, https://geoschem.github.io) to predict global [$PM_{2.5}$] and isolate the contributions from fire emissions. The model is a global three-dimensional CTM operating at a horizontal resolution of 2° latitude by 2.5° longitude with 47 vertical layers extending from ground level to the mesosphere (Yan et al., 2018;David et al., 2019;Lu et al., 2019). The model incorporates MERRA-2 meteorological inputs and implements a comprehensive chemical mechanism covering HOx-NOx-VOCs-$O_3$-halogen-aerosol interactions (Mao et al., 2013). Previous studies have extensively demonstrated the effectiveness of GEOS-Chem in simulating the reasonable distribution of trace gases and aerosols at multiple spatial and temporal scales (Xu et al., 2013;Breider et al., 2014;Li et al., 2019).

Emissions from various sources, regions, and types are processed using the Harvard–NASA Emissions Component (HEMCO) module, which operates online and allows users to specify the grid, apply scaling factors, and dynamically integrate, overlay, and update emission inventories (Keller et al., 2014). In our study, we incorporated two daily fire emission inventories within the HEMCO framework, including GFED4.1s (short as GFED thereafter) and QFED2.5 (short as QFED thereafter), both of which range from 2000 to 2023 (last modified in June 26th, 2024). All simulated PM$_{2.5}$ data at 2°×2.5° from GEOS-Chem were downscaled to 0.25°×0.25° using the bilinear interpolation (Wei et al., 2021). In addition, we estimated [PM$_{2.5}$] unaffected by fires by disabling the biomass combustion inventories in GEOS-Chem.

## 2.4 The machine learning algorithm

We used the XGBoost machine learning algorithm to bias-correct the simulated all-source [PM$_{2.5}$]. XGBoost is based on the principle of gradient tree boosting (GTB) algorithms, which combine multiple imperfect decision trees (referred to as base or weak learning trees) to create a more accurate composite decision tree (Chen and Guestrin, 2016). XGBoost is designed for efficiency and speed, capable of building trees gradually and supporting customized objective functions and evaluation metrics. These features make it particularly well-suited for various regression tasks. The primary objective of this algorithm is to minimize of loss function, enhancing the model's predictive accuracy. Notably, XGBoost provides a robust and scalable solution that optimizes computational speed and reduces memory usage when training large sample datasets (Li et al., 2020). The formula for prediction is defined as follows:

$$\hat{Y} = \sum_{k=1}^{K} f_k(X) \tag{1}$$

where $\hat{Y}$ is the predicted daily [PM$_{2.5}$]; X is the input variable related to [PM$_{2.5}$], which includes simulated [PM$_{2.5}$] from GEOS-Chem, meteorological fields, and land cover data (Table S1). $K$ is the number of decision trees used in the model, and $f_k$ denotes the tree constructed to minimize the residuals left by the (k-1)$^{th}$ tree.

XGBoost implements early stopping strategies and regularization techniques within the objective function to effectively prevent overfitting. The $k^{th}$ iteration of XGBoost function ($R^k$) is defined as follows:

$$R^k = \sum_{i=1}^{t} l\left(y_i, \widehat{y}_i^{\,k}\right) + \sum_{j=1}^{k} \Omega\left(f_j\right) \tag{2}$$

where $t$ refers to the number of samples; $y_i$ represents the actual value of the $i^{th}$ sample, and $\widehat{y}_i^{\,k}$ means the predicted value of the $i^{th}$ sample after $k$ iterations. The function $l\left(y_i, \widehat{y}_i^{\,k}\right)$ is a differentiable loss function used to measure the discrepancy between $y_i$ and $\widehat{y}_i^{\,k}$. The regularization term $\Omega\left(f_j\right)$ includes the complexity of the amount of the tree $f_j$, such as the number of nodes and the weights assigned to each node (Ma et al., 2020). In addition, XGBoost employs a second-order Taylor expansion for the loss function, enhancing the precision of the model's error assessment and consequently improving the accuracy of predictions (Wong et al., 2021).

In this study, we used meteorological data, land cover information, and simulated all-source [PM$_{2.5}$] from the GEOS-Chem model (Table S1) to develop XGBoost models. These gridded input data were interpolated to monitoring sites, and the site-level PM$_{2.5}$ measurements was used as the predictand. Due to the substantial data volume, we trained the XGBoost model on a year-by-year basis using available measurements and modeling data from the corresponding years. For each year, 80% of observational records were randomly selected to train the XGBoost model, while the remaining 20% were used as independent samples for validations. The developed and validated machine learning models were then used to derive global gridded [PM$_{2.5}$], using meteorological reanalyses, land cover data, as well as PM$_{2.5}$ outputs from GEOS-Chem models, at the resolution of 0.25°×0.25° on a daily base. We estimated fire-emitted [PM$_{2.5}$] by applying the simulated fire-to-all ratio of [PM$_{2.5}$] to the XGB-adjusted all-source [PM$_{2.5}$] following the same approach by Xu et al. (2023):

$$[PM_{2.5}]_{fire} = \frac{[PM_{2.5}]_{all}^{GC} - [PM_{2.5}]_{nofire}^{GC}}{[PM_{2.5}]_{all}^{GC}} \times [PM_{2.5}]_{all}^{XBG} \tag{3}$$

## 3 Results

### 3.1 Bias-correction and validation of all-source [PM$_{2.5}$]

Figure 1 shows the locations of monitoring sites and the corresponding [PM$_{2.5}$] in 2022. High levels of PM$_{2.5}$ are observed in Asia, especially over India and East Asia, where large anthropogenic emissions locate. Predicted all-source [PM$_{2.5}$] by the XGBoost model, which implements GEOS-Chem simulations considering GFED fire emissions, in general captures the observed spatial pattern with the determination coefficient (R$^2$) of 0.82, a root-mean-square error (RMSE) of 9.22 μg m$^{-3}$, and a normalized mean bias (NMB) of 0.71%, respectively (Fig. 1b). Similarly, the XGBoost model demonstrates good performance when applied to GEOS-Chem simulations driven with QFED inventory, as shown in Fig. S2 for the results of 2022.

Figure S3 presents the top ten most crucial features identified by the XGBoost model for predicting all-source [PM$_{2.5}$] using two different fire inventories in 2022. As expected, the data simulated by the GEOS-Chem model consistently rank as the most important feature. This is followed by meteorological variables such 10-meter wind speed, surface pressure, and boundary layer height, which have a significant influence on [PM$_{2.5}$] variations (Wei et al., 2019). Although there are variations in the importance scores of the top ten features between the two inventories, it is evident that meteorological data has a more pronounced impact on [PM$_{2.5}$] compared to land use data.

Following the same protocol, we developed machine learning models using XGBoost method for other years as well. Each year's model featured distinct parameterization schemes, and we utilized a 10-fold cross-validation (CV) method to verify the robustness of these models. The statistical indicators, including CV score R$^2$, overall R$^2$, and RMSE for the XGBoost procedures over 24 years are displayed in Fig. S4. The R$^2$ of CV validation remained above 0.85 throughout the study period, demonstrating that the XGBoost model effectively bias-corrected the predicted all-source PM$_{2.5}$ with reasonable spatial coverage and temporal stability. It should be noted that the R$^2$ gradually decreased after 2012, likely due to the rapid growth in data volume, which may have weakened the correlations (Perry and Dickson, 2018).

## 3.2 Development and validation of fire [PM$_{2.5}$]

We compare the all-source [PM$_{2.5}$] with ($[PM_{2.5}]_{all}^{XBG}$) and without ($[PM_{2.5}]_{all}^{GC}$) the XGBoost adjustment. Averaged for 24 years, the original simulations exhibit high [PM$_{2.5}$] in North Africa, India, and East Asia, and relatively high values in eastern U.S. and central Europe (Fig. 2a). With the bias-correction, those hotspots are either weakened or shrink (Fig. 2c). Specially, [PM$_{2.5}$] decreases by 28.8% in North Africa, 12.3% in India, 41.7% in East Asia, 27.5% in eastern U.S., and 35.5% in central Europe (Table 1). In contrast, the adjusted [PM$_{2.5}$] tends to increase over the regions with limited anthropogenic perturbations, such as boreal forest, Tibetan Plateau, Australian desert and so on (Fig. 2e). Such discrepancies suggests that the GEOS-Chem model generally underestimates pollution levels in pristine regions (Protonotariou et al., 2010;Kim et al., 2015) and overestimates them in areas with dense pollution (Lei et al., 2021). Similar changes in [PM$_{2.5}$] are found for the simulations using QFED (Fig. 2b, d and f) fire inventories. It worths noting that changes of [PM$_{2.5}$] varied significantly over some regions among different fire inventories. For example, the bias-corrected [PM$_{2.5}$] decreases by 19.2% in India and 33.6% in eastern U.S. with QFED (Table 1), much larger than those with GFED, suggesting that differences in fire inventories result in certain discrepancies in the regional [PM$_{2.5}$].

We derive fire-sourced [PM$_{2.5}$] using Equation (3) and validate it for typical fire events during 2018-2022 (Fig. 3). Fire carbon emissions from GFED are used to pinpoint the accurate fire locations for these events (Table S2). The fire-affected sites are determined if their back trajectory cross the fire locations within the three days after occurrence. For example, 55 sites in Canada exhibited abrupt enhancement of [PM$_{2.5}$] more than 5 times above ordinary level around August 15[th], 2023. The back trajectory of these sites aligned with the large fire emissions at the western coast during August 13-15 (Fig. S5). Similarly, 94 sites along the eastern coast of Australia were affected by the fire plume transport during December 8-10, 2019 (Fig. S6). Averaged for these 12 events, the correlation coefficient (R) of [PM$_{2.5}$] between observations and simulations increases from 0.16±0.37 without fire emissions to 0.58±0.29 with fire emissions. The NMB is improved from -53.17±25.50% without fire emissions to 10.68±24.96% with fire emissions during the correspondent fire periods. A similar improvement of [PM$_{2.5}$] is achieved with QFED emission inventory for these fire events (Fig. S7). We further compare the fire-sourced [PM$_{2.5}$] data with the estimates by Childs et al. (2022) in the U.S. (Fig.

4). Our estimates show reasonable performance, with correlation coefficients of 0.68 (0.6) and RMSE of 2.79 (2.71) $\mu g \ m^{-3}$ using the GFED (QFED) inventory. However, fire-sourced [PM$_{2.5}$] form GFED is overall lower than that of Childs et al. (2022) by -55.04%.

The probability density distributions of fire-sourced [PM$_{2.5}$] from the two inventories show notable differences (Fig. S8). During 2000-2023, fire [PM$_{2.5}$] from QFED is more than twice that from GFED below the 75$^{th}$ percentile, indicating that QFED predicts significantly higher [PM$_{2.5}$] for low to moderate fire events. However, this difference diminishes above the 90$^{th}$ percentile and becomes particularly constrained at the 99$^{th}$ percentile, where fire-sourced [PM$_{2.5}$] from GFED is 79.29% of that from QFED. It suggests that while both inventories yield comparable estimates for extreme fire episodes, GFED systematically underestimates emissions from smaller fires. This underestimation persists despite improvements in GFED's representation of small fires through additional implementations (van der Werf et al., 2017). Consequently, validations in the U.S. reveal substantial low values with GFED relative to previous estimates (Fig. 4), a bias that is alleviated in QFED for small to moderate fires (Fig. S9). Although both inventories perform comparably during high-emission events (Figs. 3 and S7), their estimates remain much lower than those of Childs et al. (2022) at the highest levels of fire-sourced [PM$_{2.5}$] (Fig. S9).

## 3.3 Spatiotemporal variations of fire-sourced [PM$_{2.5}$]

We examine the spatiotemporal variations of fire [PM$_{2.5}$] derived from the GFED inventory (Figs. 5 and S10). Averaged for 2000-2023, fire-sourced [PM$_{2.5}$] shows strong spatial heterogeneity with the highest concentrations in central Africa (CAF) and secondary hotspots in South America (SA), Southeast Asia (SEA), North America (NA), and Siberia (SB). An upward trend in fire-related [PM$_{2.5}$] is found in Siberia and North America, while most other regions show downward trends (Fig. 5b). Predictions using the QFED inventory indicate much higher long-term average fire-related [PM$_{2.5}$] compared to GFED (Table 2), particularly in the Middle East, western Siberia, and eastern South America (Fig. 5c and 5e). The decreasing trend in fire [PM$_{2.5}$] predicted by QFED is even more pronounced than that predicted by GFED in fire-prone regions such as western Siberia, South America, and Australia (Fig. 5d). In contrast, a positive trend is predicted by QFED in eastern China and Europe,

where wildfires are typically limited due to the dense population (Bistinas et al., 2014;Knorr et al., 2014).

These differences in fire-sourced [$PM_{2.5}$] are mainly due to the discrepancies in fire inventories. In global fire-prone regions, organic carbon (OC) emissions from fires are 51.08-65.18% lower in the GFED inventory compared to the QFED inventory (Fig. 6a). Consequently, the global average fire-sourced [$PM_{2.5}$] is estimated at 2.04 μg m$^{-3}$ with GFED, nearly half of the 3.96 μg m$^{-3}$ estimated with QFED (Table 2). Moreover, fire emission trends in QFED tend to be more negative or less positive than in GFED (Fig. 6b), leading to stronger negative trends in fire-sourced [$PM_{2.5}$] derived from QFED (Fig. 6d). For both inventories, simulated fire [$PM_{2.5}$] trends are more negative than the corresponding emission trends, likely due to climatic or chemical conditions that enhance pollutant removal. For example, in North America, increased atmospheric oxidant levels (e.g., increased OH and $O_3$) and changes in boundary layer height over the past two decades may have offset rising fire emissions by accelerating aerosol aging and modifying vertical mixing (Heilman et al., 2014;Zhou et al., 2019). In Siberia, the positive trend in GFED emissions is not fully reflected in fire-sourced [$PM_{2.5}$], likely due to concurrent increases in rainfall and deposition efficiency that enhance particulate scavenging (Konovalov et al., 2024).

On the long-term mean basis, fire-related [$PM_{2.5}$] is significantly higher in the tropics than in boreal regions (Fig. 5), primarily due to the high fire emissions in central Africa (Fig. 6). However, during extreme events, fire-sourced [$PM_{2.5}$] can reach comparable levels at both low and high latitudes (Fig. 7). For instance, an unprecedented fire event over Canada in 2023 resulted in a regional hotspot exceeding 30 μg m$^{-3}$, surpassing the maximum value of ~20 μg m$^{-3}$ in central Africa. Similarly, the extreme Siberian fires in 2019 significantly elevated local [$PM_{2.5}$] and resulted in air pollution levels comparable to those in South America. From 2000 to 2023, the ratio of maximum to mean fire [$PM_{2.5}$] peaked at values exceeding 4 around 60$^{o}$ in both hemispheres, gradually decreasing to 2 at lower latitudes (Fig. 7a). In the vast tropical areas, fires are primarily driven by anthropogenic activities (Ward et al., 2018;Marques et al., 2021), leading to relatively stable emissions from year to year. In contrast, most biomass burning in boreal regions is caused by wildfires, which are less inhibited by human activities. These uncontrolled fire episodes, combined with the huge carbon storage in boreal forests, result in

tremendous emissions in specific years, significantly affecting air quality, climate system, and ecosystem functions at high latitudes in the Northern Hemisphere. It worths noting that fire-sourced [PM$_{2.5}$] shows lower extreme values in QFED (Fig. S11) compared to those in GFED (Fig. 7) over Canada, though the mean fire [PM$_{2.5}$] is much higher associated with the former inventory (Fig. 6).

Extreme fire episodes pose significant threats to public health. The percentage of days and land
grids with fire-sourced [PM$_{2.5}$] exceeding the World Health Organization's air quality standard of 15 μg m$^3$ showed a global decreasing trend of -0.03% yr$^{-1}$ (Fig. 8a). Regionally, an increase of 0.04% yr$^{-1}$ was found in North America, driven by the 2023 Canadian fire episode, though this change was not statistically significant. In other regions, the exposure risk to high levels of fire PM$_{2.5}$ declines, with the most notable declines of -0.22% yr$^{-1}$ in South America and -0.13% yr$^{-1}$ in Africa. While extreme fire
[PM$_{2.5}$] in general decreased, a turning point occurred in 2017, with more pronounced fire events thereafter. To better understand recent trends, we examined changes in fire-sourced [PM$_{2.5}$] during the past few years. Relative to 2000-2019, fire [PM$_{2.5}$] decreases across nearly all latitudes from 2020 to 2023 for both inventories (Fig. 9). Regionally, hotspots of increased fire [PM$_{2.5}$] could be found in North America, due to the 2023 Canadian fires, and in the Amazon, due to the 2022 Brazilian fires.
Additionally, fire [PM$_{2.5}$] levels increased in central Africa, northern India, and the Indo-China Peninsula, where human-induced agricultural burning is prevalent (van der Werf et al., 2017).

## 4 Conclusions and discussion

We developed a global high-resolution dataset of fire-sourced PM$_{2.5}$ concentrations for the period
of 2000-2023, using a chemical transport model driven with two fire inventories. A machine learning algorithm was applied to correct biases in the simulated [PM$_{2.5}$] from the GEOS-Chem model, and to enhance the spatial resolution of data product. Validations demonstrated its high accuracy in capturing all-sourced [PM$_{2.5}$] across more than 9000 global sites and the fire-sourced [PM$_{2.5}$] for typical fire events. Though with some discrepancies, fire-sourced [PM$_{2.5}$] from the two inventories displayed a
consistent spatial pattern, with high levels of fire-related air pollution in tropics and relatively lower concentrations at middle to high latitudes. They also exhibited significant global declines in fire-sourced [PM$_{2.5}$] over time, with the most pronounced decreases occurring in tropical regions. In contrast, fire

episodes in boreal regions led to stronger enhancement of [PM$_{2.5}$] compared to those in the tropics, due to the larger fuel loads of northern forests and the uncontrolled scale of fires in these areas.

Recent advancements in ground monitoring networks and satellite observation systems have led to the development of high-resolution, long-term benchmark datasets for air pollutants (Gui et al., 2020;Song et al., 2022;Xiao et al., 2022;Wang et al., 2023;Wei et al., 2023). However, these datasets typically retrieve total amount of PM$_{2.5}$ without isolating the concentrations specifically caused by fire events. To accurately derive fire-related [PM$_{2.5}$], it is crucial to firstly estimate PM$_{2.5}$ concentrations that

are unaffected by fires. Some studies have identified fire-affected sites using satellite imagery and then obtained non-fire PM$_{2.5}$ either by taking the median [PM$_{2.5}$] during non-fire seasons or by using data at nearby sites outside the influence of fire plumes (O'Dell et al., 2019;Delp and Singer, 2020;Burke et al., 2023). This approach, however, depends heavily on the accuracy of high-frequency fire tracking systems to correctly identify fire periods and affected areas. Additionally, it may introduce biases due to

not accounting for baseline [PM$_{2.5}$] differences across various locations or periods. In our study, we performed sensitivity experiments using a CTM, where fire emissions were selectively activated or deactivated. This allowed us to more accurately quantify the changes of [PM$_{2.5}$] attributable to fire emissions. Furthermore, we employed machine-learning adjustments to minimize biases inherent in chemical models, thereby improving the accuracy and resolution of the derived fire-related [PM$_{2.5}$].

We employed a similar approach to Xu et al. (2023) but incorporated new datasets and perspectives. First, we used global observed PM$_{2.5}$ concentrations from 9541 monitoring sites, significantly more than the 5661 stations used in Xu et al. (2023). The expansion of ground-based stations, particularly in fire-prone regions such as Africa and South America, strengthens the foundation for model training and data validation. Second, we applied two different fire emission inventories. Comparisons showed that fire

[PM$_{2.5}$] estimates from these inventories were consistent during extreme fire episodes (Figs 3 and S7). However, for low to moderate fire emissions, fire [PM$_{2.5}$] from GFED was much lower than that from QFED (Fig. S8), suggesting that global population exposure to fire-related air pollution may have been underestimated in Xu et al. (2023) due to the application of GFED. Third, we extended the ending simulation year from 2019 to 2023, capturing an additional four years that included unprecedent fire

events, such as the 2023 Canadian fires and the 2022 Brazilian fires. These events provide valuable data

for assessing population exposure and associated health impacts. Fourth, we found a global decreasing trend in fire [$PM_{2.5}$] during 2000-2023, which contrasts with the increasing trend reported in Xu et al. (2023). This discrepancy may stem from differences in machine learning approaches (random forest vs. XGBoost in this study), pollution definitions (population-weighted vs. non-weighted), and observational

datasets. Despite these differences, both studies identified a turning point in 2017, after which global fire [$PM_{2.5}$] began to increase, with the most pronounced rise observed in boreal regions.

The two datasets derived from different inventories showed discrepancies in both the long-term mean and trend of fire-sourced [$PM_{2.5}$] (Fig. 5). In general, fire-related [$PM_{2.5}$] is much higher when using the QFED inventory compared to GFED, but the long-term trend is more negative with QFED. As

expected, these discrepancies can be attributed to differences in the underlying fire emission inventories (Fig. 6), which stem from variations in their estimation methods, data sources, emission factors, and so on (Kaiser et al., 2012;Larkin et al., 2014;Jin et al., 2023). For example, QFED adjusts emission factors based on aerosol optical depth from MODIS (Petrenko et al., 2012;Li et al., 2022), resulting in significantly higher emissions in some regions compared to GFED. In contrast, GFED relies on burning

pixels and changes in surface reflectance identified during satellite overpasses under relatively cloud-free conditions, which may lead to underestimating burned areas especially for some small fires (Pan et al., 2020). Further validations showed that all-source [$PM_{2.5}$] using GFED yielded an R value of $0.58\pm0.29$ and an NMB of $10.68\pm24.96\%$ averaged for the 12 fire episodes (Fig. 3). Slightly improved statistical metrics were achieved using QFED, with an R value of $0.63\pm0.26$ and an NMB of $6.56\pm27.61\%$

for the same events (Fig. S7). However, these differences are too minor to conclusively determine which dataset provides a better estimate of fire-sourced [$PM_{2.5}$]. Fire-sourced [$PM_{2.5}$] is generally lower in the GFED dataset compared to QFED; exceptions exist, such as the 2023 Canadian fires, in which fire-sourced [$PM_{2.5}$] from GFED (Fig. 7) was significantly higher than that from QFED (Fig. S11). Therefore, we recommend using the average of fire-sourced [$PM_{2.5}$] from both inventories to indicate

the mean state, while using their difference as the range of uncertainties associated with fire-related air pollutants.

There are some uncertainties and limitations in our study. First, the $PM_{2.5}$ observations used for machine learning lack broad spatial coverage. Although we gathered data from thousands of monitoring

sites worldwide, most of them are located in the middle to high latitudes of the Northern Hemisphere. PM$_{2.5}$ records are still limited in the fire-prone regions, such as central Africa, which are usually wildland areas far away from populated regions. This uneven distribution of monitoring sites may introduce some biases in the derived all-source [PM$_{2.5}$] estimates and the subsequent contributions by fire emissions. Second, we used only one machine learning method for data training. In the preliminary stages, we compared the effectiveness of three different machine learning approaches for correcting biases in simulated [PM$_{2.5}$]. We found that XGBoost algorithm outperformed the other two methods, Random Forest and Deep Neural Networks, showing better statistical metrics against observations (not shown). Although we chose XGBoost for the final analyses, further investigation into results from other machine learning algorithms is warranted to reduce uncertainties inherent in data-driven methods. Third, biases in the [PM$_{2.5}$] simulated by the GC model may significantly affect the accuracy of machine learning. Predicting air pollutants involves uncertainties due to variations in meteorological forcing, chemical and physical schemes, initial and boundary conditions, and so on. For example, Qiu et al. (2024) found that the GC model significantly overestimated [PM$_{2.5}$] during extreme wildfire events in 2020 over the western U.S. In contrast, our derived fire [PM$_{2.5}$], using the same GFED inventory, is much lower than the estimates of Childs et al. (2022) for low to median fire events (Fig. 4). These findings suggest that incorporating more validated fire inventories and/or chemical models is necessary to better quantify the uncertainties in derived air pollutant concentrations.

Despite the limitations mentioned, our study presents a significant advancement in the development of global daily fire-sourced [PM$_{2.5}$] datasets, featuring the most up-to-date, fine spatial resolution, and covering the longest time period available. By integrating a chemical model with a machine learning approach, we have effectively isolated the impact of fire emissions on ground-level [PM$_{2.5}$], while also addressing and reducing modeling biases. This methodology allows for a more accurate representation of fire-related air pollution. Furthermore, we provide results derived from two different emission inventories, offering a comparison that highlights the uncertainties associated with varying emission estimates. The dataset we have constructed is not only a novel contribution to the field but also holds significant value for future research. It can serve as a critical input for studies examining the climatic, ecological, and epidemiological impacts of air pollutants from global fires. The insights

gained from this dataset can inform policy decisions, improve public health strategies, and enhance our understanding of the broader environmental effects of wildfire emissions.

**Data availability.** The daily fire-sourced [PM$_{2.5}$] from GFED4.1s and QFED2.5 for 2018-2022 are available at https://doi.org/10.5281/zenodo.14934127 (Hu et al., 2024a). The full dataset will be publicly available after the publication of this study.

**Author contributions.** XY conceived the study. XY, CT and YH designed the research. YH performed
the model runs, completed data analysis, and wrote the draft. XY reviewed and edited the manuscript. YL, YC, RX, and YG helped with methods. All authors contributed to the discussion of the results and to the finalization of the paper.

**Financial support.** This study was jointly supported by the National Key Research and Development
Program of China (no. 2023YFF0805402), National Natural Science Foundation of China (no. 42405107), the Natural Science Foundation of Jiangsu Province (no. BK20240715), and Jiangsu Funding Program for Excellent Postdoctoral Talent (no. 2023ZB113).

**Competing interests.** The contact author has declared that none of the authors has any competing
interests.

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

610

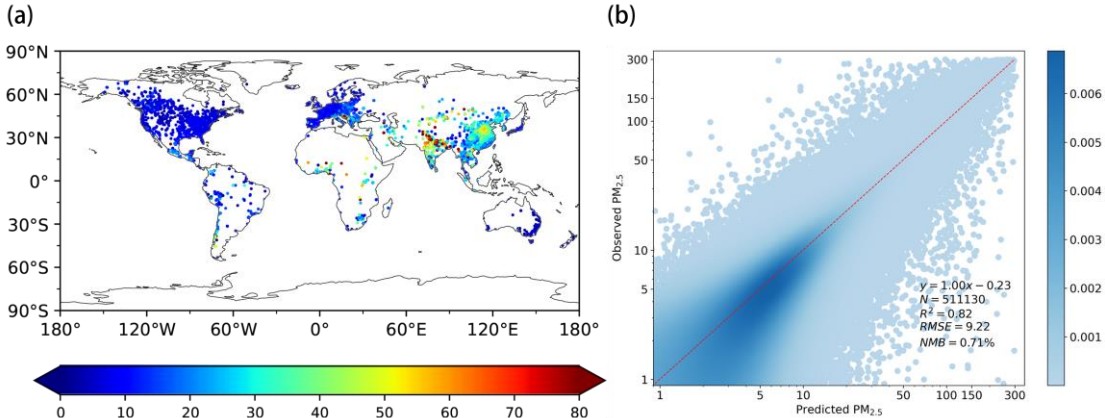

**Figure 1.** Observed PM$_{2.5}$ concentrations and their comparisons with predictions made by the XGBoost model. Panel (a)
presents the annual mean PM$_{2.5}$ concentrations ([PM$_{2.5}$], units: μg m$^{-3}$) at 9541 monitoring sites in 2022. Panel (b) shows
daily PM$_{2.5}$ concentrations predicted by the GEOS-Chem model, adjusted using the XGBoost approach, and compared with
validation subsets of observations in 2022. The GEOS-Chem simulations incorporate emissions from both anthropogenic
sources and the Global Fire Emissions Database version 4.1s. Colors in (b) represent data frequency, and the red dashed line
indicates the linear regression. Validation metrics, including the sample size (N, 20% of total observational records),
regression equation, determination coefficient (R$^2$), root-mean-square error (RMSE), and normalized mean bias (NMB), are
provided. GEOS-Chem simulations using QFED inventory for 2022 are shown in Fig. S2.

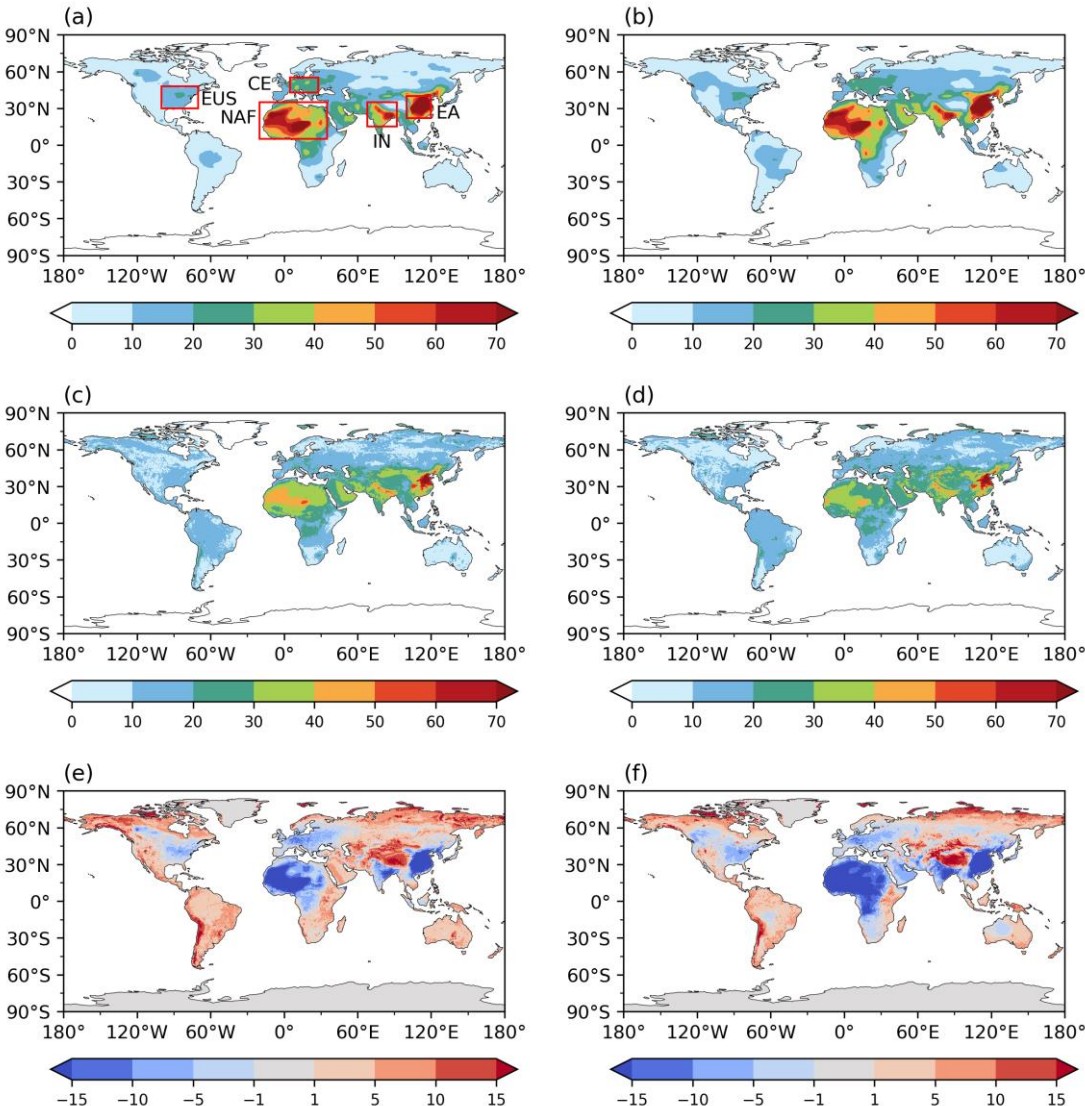

**Figure 2.** Annual mean all-source [PM$_{2.5}$] for 2000-2023 from the original GEOS-Chem simulations at 0.25°×0.25° resolution derived using (a) GFED and (b) QFED inventories, as well as (c, d) bias-corrected estimations using the XGBoost approach at the same resolution. The difference between the original and bias-corrected [PM$_{2.5}$] is shown in (e, f).

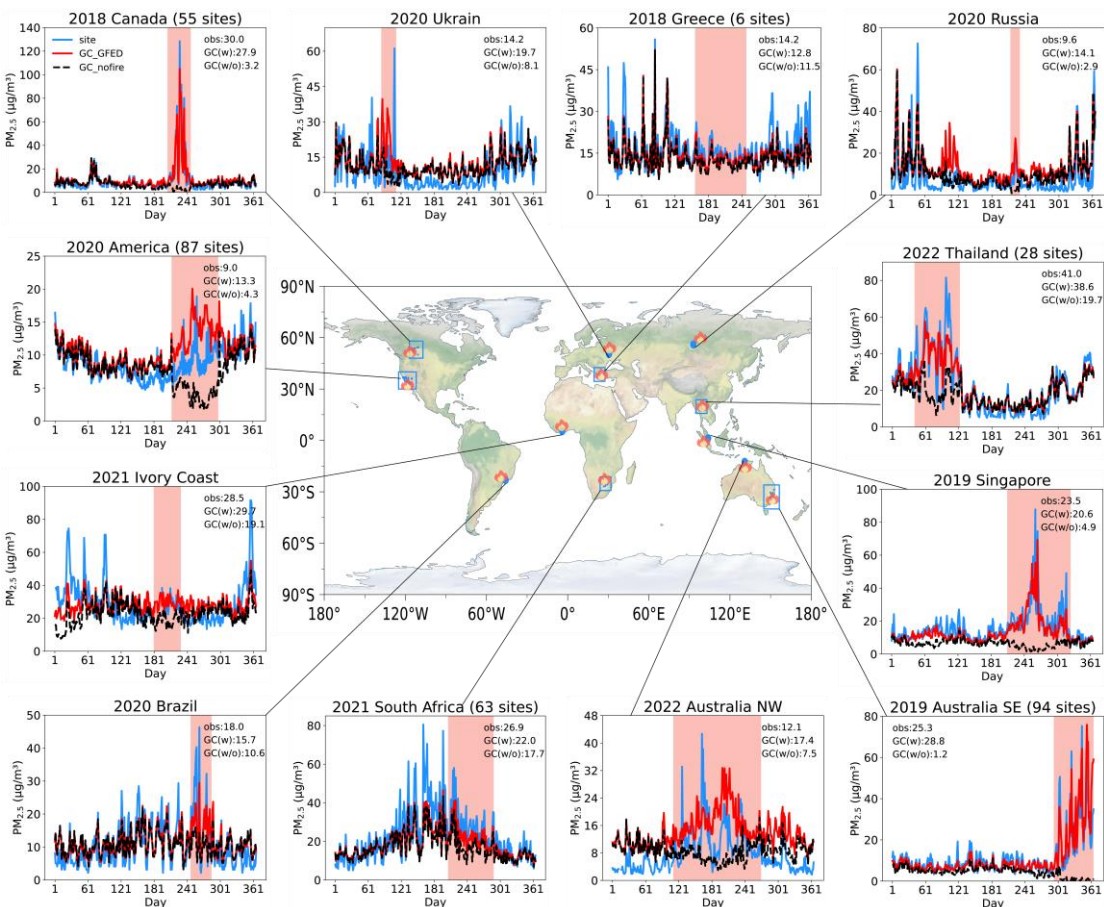

**Figure 3.** Comparisons of [PM$_{2.5}$] between observations (blue) and estimations with (red) and without (black) fire emissions for 12 incidences during 2018-2022. The estimations are performed using the GEOS-Chem model driven with fire emissions from the GFED inventory and bias-corrected with the XGBoost approach. Blue boxes (representing multiple sites) or points (representing single sites) on the map indicate the locations of air quality monitoring sites affected by nearby fire plumes. The sources of these fire episodes were determined using Lagrangian back-trajectory analysis as shown in Figs S5-S6. The observed and estimated [PM$_{2.5}$] at all sites averaged for fire periods are shown on each panel. These fire events were sourced from the Global Disaster Data Platform (https://www.gddat.cn/newGlobalWeb), which provides fire locations and the approximate start and end dates as shown in Table S2.

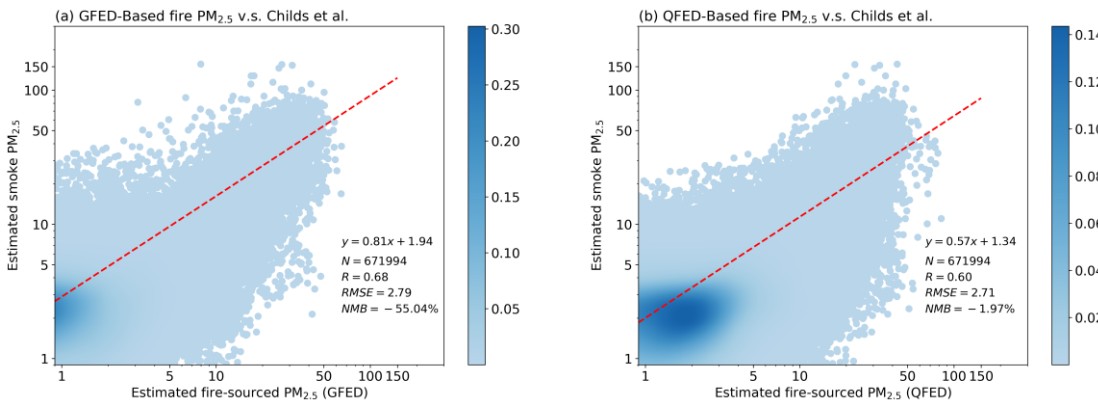

640 **Figure 4.** Comparison of fire-sourced $PM_{2.5}$ ($\mu g\ m^{-3}$) estimated using (a) GFED and (b) QFED inventories with smoke $PM_{2.5}$ observed by Childs et al. (2022) at 100156 polygons in U.S. during 2016–2019. Validation metrics of N, regression equation, $R^2$, RMSE, and NMB are calculated.

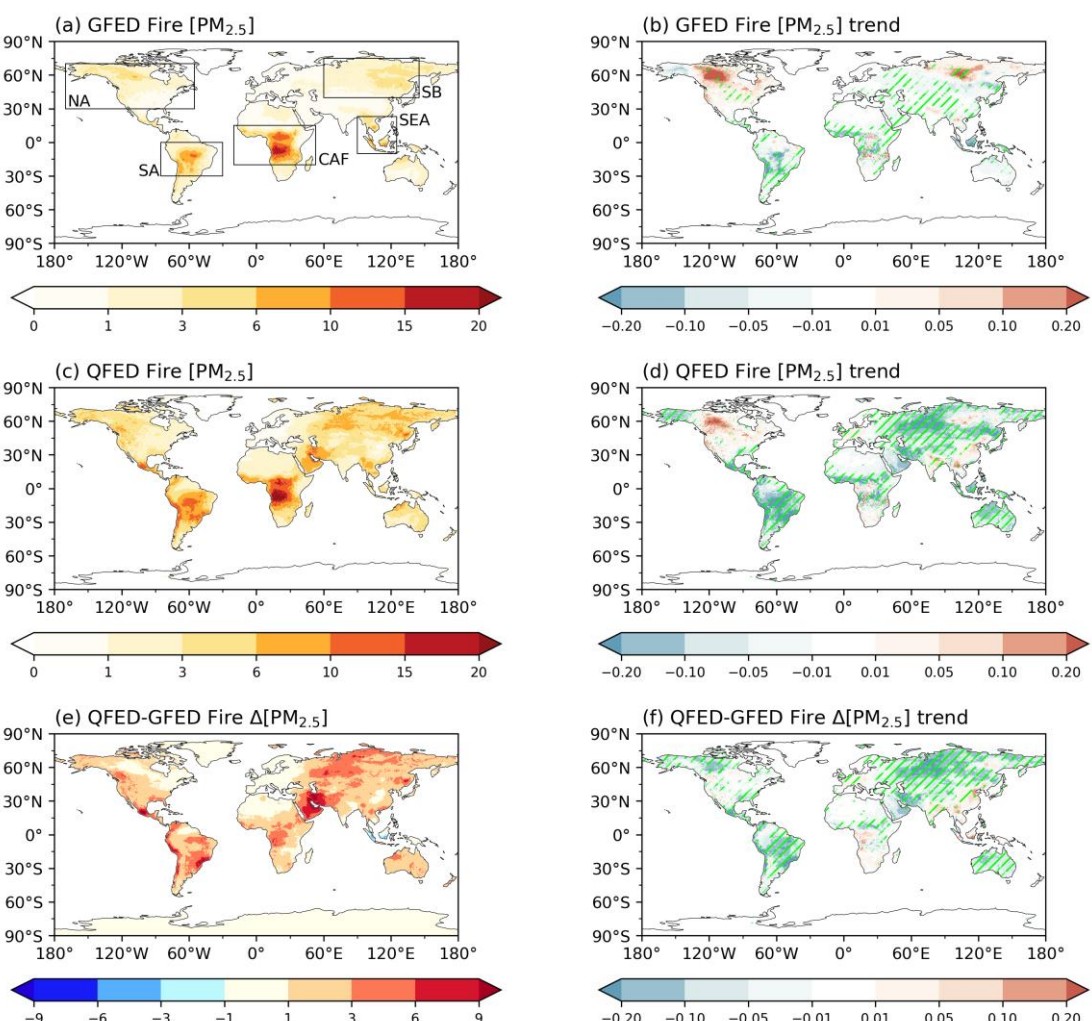

**Figure 5.** Long-term (a) mean and (b) trend of fire [PM$_{2.5}$] (µg m$^{-3}$) derived using the GFED inventory for 2000-2023. The box regions in (a) indicate areas used for comparing differences between two inventories. Panels (c) and (d) display the same information as (a) and (b), but for fire [PM$_{2.5}$] from QFED inventory. The differences in fire [PM$_{2.5}$] ($\Delta$[PM$_{2.5}$]) between the two inventories are presented for the long-term (e) mean and (f) trend during 2000-2023. Green slashes indicate areas with significant ($p < 0.05$) changes. The $p$ values of these trend are shown in Fig. S10.

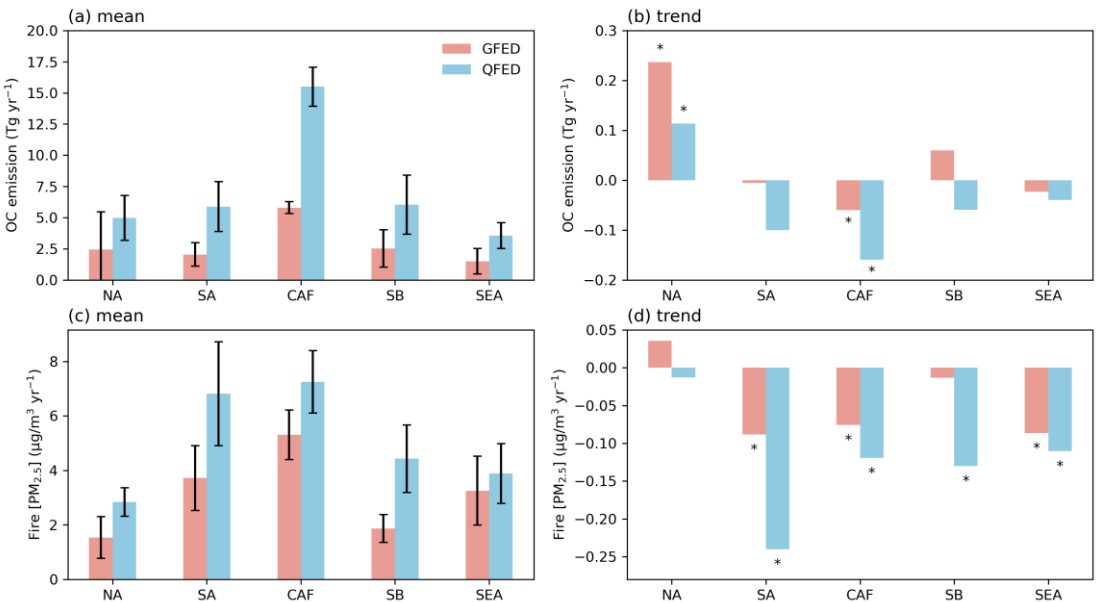

**Figure 6.** The mean (a) and trend (b) of OC emissions in high fire-prone regions indicated by the GFED (red) and QFED (blue) inventories. Panels (c) and (d) display the fire-sourced [PM$_{2.5}$] predicted using these two inventories. Errorbars represent one standard deviation for the year-to-year variations, and an asterisk denotes areas with significant ($p < 0.05$) trends. Domain of the labelled regions is shown on Fig. 5a.

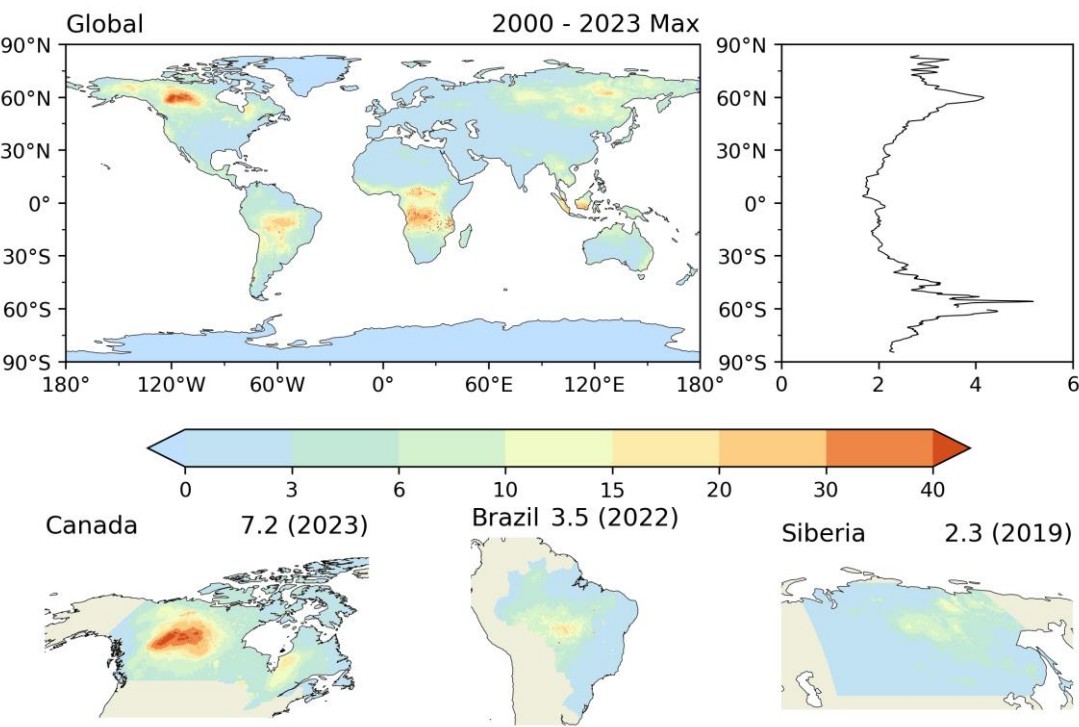

**Figure 7.** The global maximum of fire [PM$_{2.5}$] (μg m$^{-3}$) from 2000 to 2023 derived using the GFED inventory, along with fire [PM$_{2.5}$] during years of high wildfire emissions in various regions. For each grid on the global map, the maximum fire-sourced [PM$_{2.5}$] during 2000-2023 is shown. The ratios between zonal maximum and mean values are shown alongside the panel. For the three regions, numbers before the parentheses represent the mean fire [PM$_{2.5}$] averaged for the affected countries or regions in that specific year.

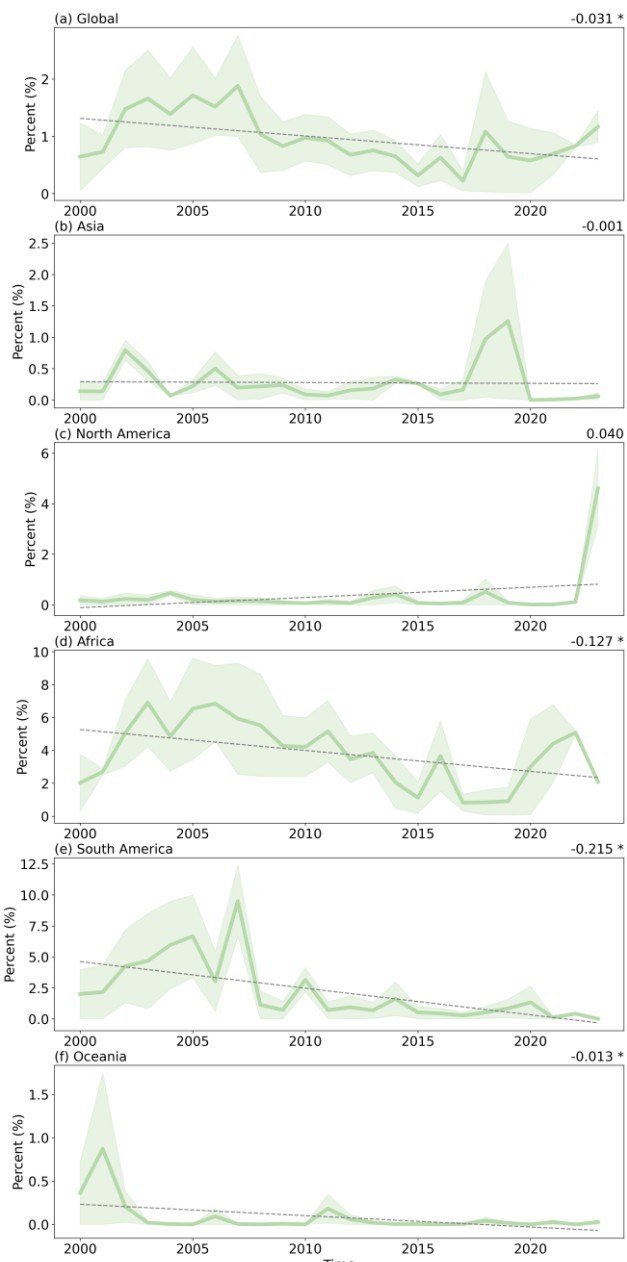

**Figure 8.** Annual percentage of days and land grids with fire-sourced [$PM_{2.5}$] exceeding 15 μg m$^3$ in (a) Global, (b) Asia, (c) North America, (d) Africa, (e) South America and (f) Oceania for 2000-2023. The average estimates from GFED and QFED are shown as bold lines, with shadings indicating their range. Regional trends are displayed on the top right of each panel, with an asterisk denoting significant ($p < 0.05$) changes.

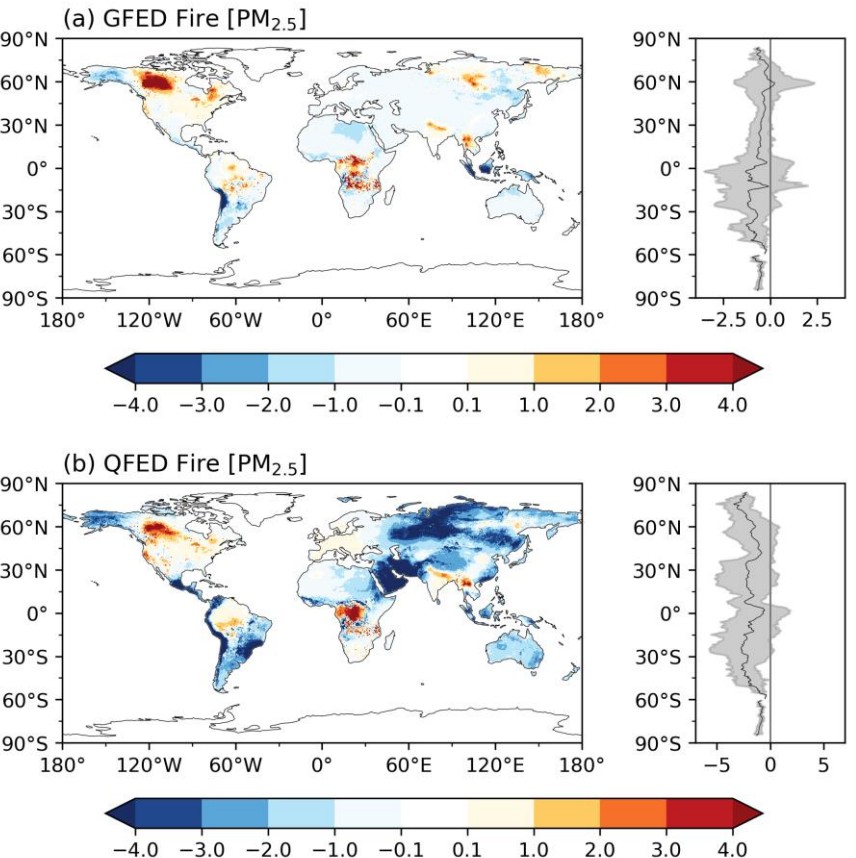

**Figure 9.** Differences in estimated fire-sourced PM$_{2.5}$ (µg m$^{-3}$) between 2020-2023 and 2000-2019 derived using (a) GFED and (b) QFED inventories. The zonal averages and one standard deviation are shown alongside each panel.

**Table 1.** Mean PM$_{2.5}$ before and after bias-correction in selected regions averaged for 2000-2023

|      |      | EUS | CE | NAF | IN | EA |
|------|------|-----|-----|-----|-----|-----|
|      | GC   | 14.38±3.39 | 24.74±3.67 | 45.97±16.46 | 33.73±16.36 | 68.68±33.41 |
| GFED | XGB  | 10.42±2.16 | 15.94±3.33 | 32.74±8.08 | 29.59±8.07 | 40.08±16.73 |
|      | Diff | 3.96±3.13 | 8.79±3.85 | 13.22±10.33 | 4.14±10.54 | 28.61±20.14 |
|      | GC   | 16.55±3.44 | 23.56±3.25 | 47.56±15.82 | 35.60±16.30 | 70.57±32.36 |
| QFED | XGB  | 10.99±2.58 | 16.10±3.20 | 27.09±7.20 | 28.78±7.07 | 39.00±16.95 |
|      | Diff | 5.56±3.38 | 7.46±3.81 | 20.47±10.57 | 6.82±12.90 | 31.57±19.79 |

**Table 2.** The mean fire-induced [PM$_{2.5}$] in selected regions averaged for 2000-2023

|      | NA | SA | CAF | SB | SEA | Global |
|------|-----|-----|-----|-----|-----|--------|
| GFED | 1.53±0.99 | 3.72±2.43 | 5.31±4.28 | 1.87±1.08 | 3.25±1.72 | 2.04±2.33 |
| QFED | 2.84±1.30 | 6.81±2.77 | 7.25±4.99 | 4.43±1.65 | 3.88±1.62 | 3.96±3.01 |
