# Peer review of "Figure S1. Comparison of observed and interpolated PM2.5 concentrations ([PM2.5]) in China from 2015 to 2022. Panel (a) displays annual mean [PM2.5] ( $\mu\text{g m}^{-3}$ ) at 1822 monitoring sites. Panel (b) shows interpolated annual mean [PM2.5] fro"

_Earth System Science Data, 2024_

## Author Comment (AC1)

We are grateful to the editor and referee for their time and energy in providing helpful comments and guidance that have improved the manuscript. In this document, we describe how we have addressed the reviewer's comments. Referee comments are shown in black italics and author responses are shown in blue regular text.

*Reviewer #1:*

*I cannot support the acceptance of this paper at its present form due to the following major concerns*

➢ We made substantial revisions following your comments. We hope this version of paper have answered your concerns.

*Major concerns:*

*1) The method and interpretation are very similar (nearly identical) to a recent publication (Xu et al., Nature, s41586-023-06398-6, 2023). There are essentially no new developments after I examined the whole paper, except for the slightly extended time coverage (by including three additional years). If the authors intended to revise the manuscript, they should extensively discuss how and why their method and results are different from the Xu et al. study.*

➢ In the revised version, we included additional analyses and explicitly explained how our study built upon and extended findings of Xu et al. (2023):

(1) We analyzed the long-term trend of fire-sourced [$PM_{2.5}$] exceeding the WHO health standard (Figure 8) and examined recent changes in fire-related air pollutants over the past four years beyond 2020 (Figure 9). These new results enhanced the novelty of our study.

(2) In the revised discussion, we explicitly outlined how our study made further progresses compared to Xu et al. (2023):

"We employed a similar approach to Xu et al. (2023) but incorporated new datasets and perspectives. First, we used global observed $PM_{2.5}$ concentrations from 9541 monitoring sites, significantly more than the 5661 stations used in Xu et al. (2023). The expansion of ground-based stations, particularly in fire-prone regions such as Africa and South America, strengthens the foundation for model training and data validation. Second, we applied two different fire emission inventories. Comparisons showed that fire [$PM_{2.5}$] estimates from these inventories were consistent during extreme fire episodes (Figs 3 and S7). However, for low to moderate fire emissions, fire [$PM_{2.5}$] from GFED was much lower than that from QFED (Fig. S8), suggesting that global population

exposure to fire-related air pollution may have been underestimated in Xu et al. (2023) due to the application of GFED. Third, we extended the ending simulation year from 2019 to 2023, capturing an additional four years that included unprecedent fire events, such as the 2023 Canadian fires and the 2022 Brazilian fires. These events provide valuable data for assessing population exposure and associated health impacts. Fourth, we found a global decreasing trend in fire [$PM_{2.5}$] during 2000-2023, which contrasts with the increasing trend reported in Xu et al. (2023). This discrepancy may stem from differences in machine learning approaches (random forest vs. XGBoost in this study), pollution definitions (population-weighted vs. non-weighted), and observational datasets. Despite these differences, both studies identified a turning point in 2017, after which global fire [$PM_{2.5}$] began to increase, with the most pronounced rise observed in boreal regions." (Lines 330-346)

*2) There appears to be very limited discussion about uncertainties in the derived datasets. The Zenodo archive only presents absolute concentrations, while no information about the expected error was included in the data or discussed in the paper. Especially considering that the paper presented strong dependence of the fire-induced $PM_{2.5}$ on the specific fire inventory, what uncertainty envelope do you recommend in each of the dataset? After all, these datasets are expected to be used by the community for various applications, and such information is vital.*

➤ The inclusion of two inventories is one the major contributions of our study to the community. In the revised version, we added Figure 4 to validate the derived fire [$PM_{2.5}$] from both inventories against estimates from Childs et al. (2022), and Figure S8 to compare the differences between them at various percentiles. Along with other figures (e.g., Figs 3 vs. S7, Figs 7 vs. S9) and tables (Tables 1-2), our study provided a thorough comparison and quantification of the uncertainties in fire [$PM_{2.5}$] derived from two inventories. In the revised paper, we expanded our discussion on the causes of these uncertainties and offered recommendations on how to best use these datasets:

"The two datasets derived from different inventories showed discrepancies in both the long-term mean and trend of fire-sourced [$PM_{2.5}$] (Fig. 5). In general, fire-related [$PM_{2.5}$] is much higher when using the QFED inventory compared to GFED, but the long-term trend is more negative with QFED. As expected, these discrepancies can be attributed to differences in the underlying fire emission

inventories (Fig. 6), which stem from variations in their estimation methods, data sources, emission factors, and so on (Kaiser et al., 2012; Larkin et al., 2014; Jin et al., 2023). For example, QFED adjusts emission factors based on aerosol optical depth from MODIS (Petrenko et al., 2012; Li et al., 2022), resulting in significantly higher emissions in some regions compared to GFED. In contrast, GFED relies on burning pixels and changes in surface reflectance identified during satellite overpasses under relatively cloud-free conditions, which may lead to underestimating burned areas especially for some small fires (Pan et al., 2020). Further validations showed that all-source [$PM_{2.5}$] using GFED yielded an R value of 0.58±0.29 and an NMB of 10.68±24.96% averaged for the 12 fire episodes (Fig. 3). Slightly improved statistical metrics were achieved using QFED, with an R value of 0.63±0.26 and an NMB of 6.56±27.61% for the same events (Fig. S7). However, these differences are too minor to conclusively determine which dataset provides a better estimate of fire-sourced [$PM_{2.5}$]. Fire-sourced [$PM_{2.5}$] is generally lower in the GFED dataset compared to QFED; exceptions exist, such as the 2023 Canadian fires, in which fire-sourced [$PM_{2.5}$] from GFED (Fig. 7) was significantly higher than that from QFED (Fig. S9). Therefore, we recommend using the average of fire-sourced [$PM_{2.5}$] from both inventories to indicate the mean state, while using their difference as the range of uncertainties associated with fire-related air pollutants." (Lines 347-366)

*3) The paper only provided evaluation of the total $PM_{2.5}$ using ground-based measurements, which is insufficient and partially reflected by the fact that the GFED- and QFED-derived products both agree well in terms of total $PM_{2.5}$ while fire-$PM_{2.5}$ are different systematically. Many recent products of fire-$PM_{2.5}$ have been developed in North America (e.g., 10.1021/acs.est.2c02934, 10.1038/s41586-023-06522-6). The manuscript should use these critical data sources to inter-compare with the modeled fire fraction and the final estimates of fire-$PM_{2.5}$.*

➢ Thank you for your valuable suggestions. In the revised version, we added Figure 4 and related descriptions to validate the derived fire [$PM_{2.5}$] from both inventories against estimates from Childs et al. (2022): "We further compare the fire-sourced [$PM_{2.5}$] data with the estimates by Childs et al. (2022) in the U.S. (Fig. 4). Our estimates show reasonable performance, with correlation coefficients of 0.68 (0.6) and RMSE of 2.79 (2.71) μg m$^{-3}$ using the GFED (QFED) inventory. However, fire-sourced [$PM_{2.5}$] form GFED is overall lower than that of Childs et al. (2022)

by -55.04%." (Lines 239-243)

We also added Figure S8 to compare the differences in fire [PM$_{2.5}$] between the two inventories at various percentiles. This comparison helps explain why the two datasets exhibited comparable performance for fire episodes, despite a large difference in their mean values: "The probability density distributions of fire-sourced [PM$_{2.5}$] from the two inventories show notable differences (Fig. S8). During 2000-2023, fire [PM$_{2.5}$] from QFED is more than twice that from GFED below the 75$^{th}$ percentile, indicating that QFED predicts significantly higher [PM$_{2.5}$] for low to moderate fire events. However, this difference diminishes above the 90$^{th}$ percentile and becomes particularly constrained at the 99$^{th}$ percentile, where fire-sourced [PM$_{2.5}$] from GFED is 79.29% of that from QFED. It suggests that while both inventories yield comparable estimates for extreme fire episodes, GFED systematically underestimates emissions from smaller fires. This underestimation persists despite improvements in GFED's representation of small fires through additional implementations (Van Der Werf et al., 2017). Consequently, validations in the U.S. reveal substantial low values with GFED relative to observations (Fig. 4), whereas both inventories perform comparably during high-emission fire episodes (Figs. 3 and S7)." (Lines 244-254)

[Figure]

**Figure 4.** Comparison of fire-sourced PM$_{2.5}$ (μg m$^{-3}$) estimated using (a) GFED and (b) QFED inventories with smoke PM$_{2.5}$ observed by Childs et al. (2022) at 100156 polygons in U.S. during 2016–2019. Validation metrics of N, regression equation, R$^2$, RMSE, and NMB are calculated.

[Figure]

**Figure 8.** Annual percentage of days and land grids with fire-sourced [$PM_{2.5}$] exceeding 15 µg m$^3$ in (a) Global, (b) Asia, (c) North America, (d) Africa, (e) South America and (f) Oceania for 2000-2023. The average estimates from GFED and QFED are shown as bold lines, with shadings indicating their range. Regional trends are displayed on the top right of each panel, with an asterisk denoting significant ($p < 0.05$) changes.

[Figure]

**Figure 9.** Differences in estimated fire-sourced $PM_{2.5}$ (µg m$^{-3}$) between 2020-2023 and 2000-2019 derived using (a) GFED and (b) QFED inventories. The zonal averages and one standard deviation are shown alongside each panel.

[Figure]

**Figure S8.** Comparison of daily fire-sourced [$PM_{2.5}$] at different percentiles between simulations with GFED and QFED inventories.

*Other comments:*

*1) I downloaded one example data, and found that negative values occur in occasional pixels. What are the physical meanings of them?*

➢ Fire-sourced [$PM_{2.5}$] is estimated as the difference between the simulated [$PM_{2.5}$] with all sources and that without fire emissions. In some rare case, the latter might be higher than the former due to nonlinearity in chemical reactions and dynamic transport processes. However, these negative values are generally very small in absolute terms. In the revised version, we have removed all negative values by defining them as zero, which led to minor changes in regional and global statistics. For example, the original statement in the Abstract, "Globally, the average fire-sourced [$PM_{2.5}$] is estimated to be 1.94 μg m$^{-3}$ with GFED4.1s and 3.74 μg m$^{-3}$ with QFED2.5." was changed to "Globally, the average fire-sourced [$PM_{2.5}$] is estimated to be 2.04 μg m$^{-3}$ with GFED4.1s and 3.96 μg m$^{-3}$ with QFED2.5." in the revision.

*2) Line 44, fire $PM_{2.5}$ aerosols can be larger in size than urban $PM_{2.5}$, see e.g., https://acp.copernicus.org/articles/19/6561/2019/*

➢ In the revised version, we removed this sentence to avoid inaccurate statements.

*3) Line 71-81: These uncertainties seem not narrowed in this new dataset compared to the previous studies? Even the Xu et al. 2023 study itself has indicated similar differences in the derived fire-$PM_{2.5}$ using four inventories. So what new insights/constraints have this work provided?*

➢ Xu et al. (2023) compared the uncertainties in fire [$PM_{2.5}$] from different inventories only for the year 2012. In contrast, our study provided datasets from both inventories spanning 2000-2023, enabling us to compare their spatiotemporal variations. In the revised version, we provided several new insights. For example, we conducted a more in-depth analysis of long-term changes in extreme wildfire events and discussed the underlying reasons for the differences between GFED and QFED, as follows:

"Extreme fire episodes pose significant threats to public health. The percentage of days and land grids with fire-sourced [$PM_{2.5}$] exceeding the World Health Organization's air quality standard of 15 μg m$^3$ showed a global decreasing trend of -0.03% yr$^{-1}$ (Fig. 8a). Regionally, an increase of 0.04% yr$^{-1}$ was found in North America, driven by the 2023 Canadian fire episode, though this change was not statistically significant. In other regions, the exposure risk to high levels of fire

PM$_{2.5}$ declines, with the most notable declines of -0.22% yr$^{-1}$ in South America and -0.13% yr$^{-1}$ in Africa. While extreme fire [PM$_{2.5}$] in general decreased, a turning point occurred in 2017, with more pronounced fire events thereafter. To better understand recent trends, we examined changes in fire-sourced [PM$_{2.5}$] during the past few years. Relative to 2000-2019, fire [PM$_{2.5}$] decreases across nearly all latitudes from 2020 to 2023 for both inventories (Fig. 9). Regionally, hotspots of increased fire [PM$_{2.5}$] could be found in North America, due to the 2023 Canadian fires, and in the Amazon, due to the 2022 Brazilian fires. Additionally, fire [PM$_{2.5}$] levels increased in central Africa, northern India, and the Indo-China Peninsula, where human-induced agricultural burning is prevalent (Van Der Werf et al., 2017)." (Lines 289-301)

"In general, fire-related [PM$_{2.5}$] is much higher when using the QFED inventory compared to GFED, but the long-term trend is more negative with QFED. As expected, these discrepancies can be attributed to differences in the underlying fire emission inventories (Fig. 6), which stem from variations in their estimation methods, data sources, emission factors, and so on (Kaiser et al., 2012; Larkin et al., 2014; Jin et al., 2023). For example, QFED adjusts emission factors based on aerosol optical depth from MODIS (Petrenko et al., 2012; Li et al., 2022), resulting in significantly higher emissions in some regions compared to GFED. In contrast, GFED relies on burning pixels and changes in surface reflectance identified during satellite overpasses under relatively cloud-free conditions, which may lead to underestimating burned areas especially for some small fires (Pan et al., 2020)." (Lines 348-357)

*4) Line 93-94: I do not think computational cost is a major obstacle of machine learning approach.*

➢ We clarified that the computational cost is attributed to CTM simulations instead of machine learning approach: "However, due to the high computational cost, most CTM simulations have been performed at the regional scale or driven with a single fire inventory, limiting the ability of machine learning methods to accurately constrain fire-related air pollutants on global and long-term scales." (Lines 88-90)

*5) Line 108-110: Is it necessary/critical to do this specifically for China? Many other regions also bear with incomplete time series. If the ML method is very sensitive to the availability of data over 2000-2013 in China, how uncertain are your predictions for*

*e.g., India before ~2010 when observation data is available?*

➢ We tried our best to maximize the temporal and spatial coverage of site-level data while ensuring high accuracy. The TAP data was developed using machine learning approaches that integrate multiple data sources, including ground measurements, satellite retrievals, emission inventories, chemical transport model simulations, meteorological fields, and land use data http://tapdata.org.cn/?page_id=67&lang=en). Our validations further confirmed its reliability (Fig. S1). We did not find other available and robust products to expand site-level data.

*6) Line 117-118: Please provide references of the method to convert AQI to PM$_{2.5}$.*

➢ In the revised version, we modified: "where the Air Quality Index (AQI) was converted to PM$_{2.5}$ following a standardized methodology (Benchrif et al., 2021)." (Lines 112-113)

*7) Figure 1b: It appears that log-scale color scheme is needed.*

➢ In the revised version, we modified Figure 1b and Figure S2 by using log-scale x and y axis so as to visualize the data density.

[Figure]

**Figure 1.** Observed PM$_{2.5}$ concentrations and their comparisons with predictions made by the XGBoost model. Panel (a) presents the annual mean PM$_{2.5}$ concentrations ($\mu$g m$^{-3}$) at 9541 monitoring sites in 2022. Panel (b) shows daily PM$_{2.5}$ concentrations predicted by the GEOS-Chem model, adjusted using the XGBoost approach, and compared with validation subsets of observations in 2022. The GEOS-Chem simulations incorporate emissions from both anthropogenic sources and the Global Fire Emissions Database version 4.1s. Colors in (b) represent data frequency, and the red dashed line indicates the linear regression. Validation metrics, including the sample size (N, 20% of total observational records), regression equation, determination coefficient (R$^2$), root-mean-square error (RMSE), and normalized mean bias (NMB), are provided. GEOS-Chem simulations using QFED inventory for 2022 are shown in Fig. S2.

[Figure]

**Figure S2.** Same as Fig. 1b, but for GEOS-Chem simulations using the QFED inventory.

*8) Figure 3: It looks abnormal to me that the cross-validation R2 (Panel b) values are stronger than the direct R2 (Panel c) in many years? Also, please do not use "simulated" for ML-corrected PM$_{2.5}$. Could use "estimated".*

➢ We have moved the original Figure 3 into SI as Figure S4. We have changed "simulated" to "estimated" as suggested.

Typically, 80% of the data is used for training, while the remaining 20% is set aside for validation (Bai et al., 2022). During the training process, we continuously adjust the model's parameters, using the 10-fold cross-validation $R^2$ as a measure of the model's performance (Adams et al., 2020; Wang et al., 2022). Once the 10-fold cross-validation $R^2$ reaches a sufficiently high value, we validate the model using the validation set. Consequently, the lower $R^2$ shown in Panel c, compared to the cross-validation $R^2$ in Panel b, can be attributed to the difference in the data sets. Specifically, the cross-validation $R^2$ is calculated based on the training set (i.e., the 80%), while the direct $R^2$ is computed on the validation set (i.e., the rest 20%). It is common for the cross-validation $R^2$ to be slightly higher than the direct $R^2$ (Wei et al., 2019; Song et al., 2021; Xu et al., 2023).

In the text, we explained that: "For each year, 80% of observational records were randomly selected to train the XGBoost model, while the remaining 20% were used as independent samples for validations." (Lines 178-180) In the caption of Figure S4, we clarified that: "Panels (c) and (d) display the year-to-year $R^2$ and RMSE between observed and estimated [PM$_{2.5}$] using these different fire emission inventories for independent validation samples."

*9) Figure 6: I do not understand the "green slashes". Why are they so regularly distributed?*

➢ The green slashes indicate that the trend in that region passed the significance test ($p < 0.05$). Since we masked the oceanic regions, the green dashed lines are all located within the continental regions.

**References**

Adams, M. D., Massey, F., Chastko, K., and Cupini, C.: Spatial modelling of particulate matter air pollution sensor measurements collected by community scientists while cycling, land use regression with spatial cross-validation, and applications of machine learning for data correction, Atmospheric Environment, 230, 117479, https://doi.org/10.1016/j.atmosenv.2020.117479, 2020.

Bai, K., Li, K., Ma, M., Li, K., Li, Z., Guo, J., Chang, N.-B., Tan, Z., and Han, D.: LGHAP: the Long-term Gap-free High-resolution Air Pollutant concentration dataset, derived via tensor-flow-based multimodal data fusion, Earth System Science Data, 14, 907-927, https://doi.org/10.5194/essd-14-907-2022, 2022.

Childs, M. L., Li, J., Wen, J., Heft-Neal, S., Driscoll, A., Wang, S., Gould, C. F., Qiu, M., Burney, J., and Burke, M.: Daily Local-Level Estimates of Ambient Wildfire Smoke $PM_{2.5}$ for the Contiguous US, Environmental Science & Technology, 56, 13607-13621, https://doi.org/10.1021/acs.est.2c02934, 2022.

Jin, L., Permar, W., Selimovic, V., Ketcherside, D., Yokelson, R. J., Hornbrook, R. S., Apel, E. C., Ku, I. T., Collett Jr, J. L., Sullivan, A. P., Jaffe, D. A., Pierce, J. R., Fried, A., Coggon, M. M., Gkatzelis, G. I., Warneke, C., Fischer, E. V., and Hu, L.: Constraining emissions of volatile organic compounds from western US wildfires with WE-CAN and FIREX-AQ airborne observations, Atmos. Chem. Phys., 23, 5969-5991, https://doi.org/10.5194/acp-23-5969-2023, 2023.

Kaiser, J. W., Heil, A., Andreae, M. O., Benedetti, A., Chubarova, N., Jones, L., Morcrette, J. J., Razinger, M., Schultz, M. G., Suttie, M., and van der Werf, G. R.: Biomass burning emissions estimated with a global fire assimilation system based on observed fire radiative power, Biogeosciences, 9, 527-554, https://doi.org/10.5194/bg-9-527-2012, 2012.

Larkin, N. K., Raffuse, S. M., and Strand, T. M.: Wildland fire emissions, carbon, and climate: U.S. emissions inventories, Forest Ecology and Management, 317, 61-69, https://doi.org/10.1016/j.foreco.2013.09.012, 2014.

Li, F., Zhang, X., Kondragunta, S., Lu, X., Csiszar, I., and Schmidt, C. C.: Hourly biomass burning emissions product from blended geostationary and polar-orbiting satellites for air quality forecasting applications, Remote Sensing of Environment, 281, 113237, https://doi.org/10.1016/j.rse.2022.113237, 2022.

Pan, X., Ichoku, C., Chin, M., Bian, H., Darmenov, A., Colarco, P., Ellison, L., Kucsera, T., da Silva, A., Wang, J., Oda, T., and Cui, G.: Six global biomass burning emission datasets: intercomparison and application in one global aerosol model, Atmos.

Chem. Phys., 20, 969-994, https://doi.org/10.5194/acp-20-969-2020, 2020.

Petrenko, M., Kahn, R., Chin, M., Soja, A., Kucsera, T., and Harshvardhan: The use of satellite-measured aerosol optical depth to constrain biomass burning emissions source strength in the global model GOCART, Journal of Geophysical Research: Atmospheres, 117, https://doi.org/10.1029/2012JD017870, 2012.

Song, Z., Chen, B., Huang, Y., Dong, L., and Yang, T.: Estimation of PM2.5 concentration in China using linear hybrid machine learning model, Atmos. Meas. Tech., 14, 5333-5347, https://doi.org/10.5194/amt-14-5333-2021, 2021.

van der Werf, G. R., Randerson, J. T., Giglio, L., van Leeuwen, T. T., Chen, Y., Rogers, B. M., Mu, M., van Marle, M. J. E., Morton, D. C., Collatz, G. J., Yokelson, R. J., and Kasibhatla, P. S.: Global fire emissions estimates during 1997–2016, Earth Syst. Sci. Data, 9, 697-720, https://doi.org/10.5194/essd-9-697-2017, 2017.

Wang, J., He, L., Lu, X., Zhou, L., Tang, H., Yan, Y., and Ma, W.: A full-coverage estimation of $PM_{2.5}$ concentrations using a hybrid XGBoost-WD model and WRF-simulated meteorological fields in the Yangtze River Delta Urban Agglomeration, China, Environmental Research, 203, 111799, https://doi.org/10.1016/j.envres.2021.111799, 2022.

Wei, J., Huang, W., Li, Z., Xue, W., Peng, Y., Sun, L., and Cribb, M.: Estimating 1-km-resolution $PM_{2.5}$ concentrations across China using the space-time random forest approach, Remote Sensing of Environment, 231, 111221, https://doi.org/10.1016/j.rse.2019.111221, 2019.

Xu, R., Ye, T., Yue, X., Yang, Z., Yu, W., Zhang, Y., Bell, M. L., Morawska, L., Yu, P., Zhang, Y., Wu, Y., Liu, Y., Johnston, F., Lei, Y., Abramson, M. J., Guo, Y., and Li, S.: Global population exposure to landscape fire air pollution from 2000 to 2019, Nature, 621, 521-529, https://doi.org/10.1038/s41586-023-06398-6, 2023.

---

## Author Comment (AC2)

We are grateful to the editor and referee for their time and energy in providing helpful comments and guidance that have improved the manuscript. In this document, we describe how we have addressed the reviewer's comments. Referee comments are shown in black italics and author responses are shown in blue regular text.

*Reviewer #2:*

*In this paper, Hu et al. documented an important effort of generating a global daily fire-sourced $PM_{2.5}$ dataset from 2000-2023. This dataset is derived using the following steps: 1) They use GEOS-Chem to simulate global daily all-source $PM_{2.5}$ and apply ML to de-bias the all-source $PM_{2.5}$ against surface measurements; 2) they then apply the ratio between simulated fire $PM_{2.5}$ and total $PM_{2.5}$ at the grid level to estimate the fire-sourced $PM_{2.5}$. I think this is an important effort, and the dataset generated will become a valuable asset to the academic communities across different disciplines. I am excited to see such a dataset being made publicly available to the community.*

➢ Thank you for your positive evaluations.

*With that being said, I agree with the other reviewer that the paper should do a better job of discussing its difference from the Xu et al., 2023 Nature paper, in addition to the extended temporal period (which is an important update in my opinion). In addition to a transparent and detailed discussion of the differences and contributions, I imagine the project could benefit from several potential analyses to further differentiate this paper from their prior contributions:*

*1) discuss the recent trends from 2020-2023;*

➢ Thank you for your valuable suggestion. In the revised paper, we added Figure 9 to examine recent changes in fire-related air pollutants over the past four years beyond 2020. We described it as follows: "To better understand recent trends, we examined changes in fire-sourced [$PM_{2.5}$] during the past few years. Relative to 2000-2019, fire [$PM_{2.5}$] decreases across nearly all latitudes from 2020 to 2023 for both inventories (Fig. 9). Regionally, hotspots of increased fire [$PM_{2.5}$] could be found in North America, due to the 2023 Canadian fires, and in the Amazon, due to the 2022 Brazilian fires. Additionally, fire [$PM_{2.5}$] levels increased in central Africa, northern India, and the Indo-China Peninsula, where human-induced agricultural burning is prevalent (Van Der Werf et al., 2017)." (Lines 296-301)

[Figure]

**Figure 9.** Differences in estimated fire-sourced PM$_{2.5}$ (µg m$^{-3}$) between 2020-2023 and 2000-2019 derived using (a) GFED and (b) QFED inventories. The zonal averages and one standard deviation are shown alongside each panel.

*2) discuss the differences between the GFED and QFED-based dataset (which do not seem to be the focus of their prior work);*

➢ Yes, the inclusion of two inventories is one of the major contributions of our study to the community. In the revised paper, we added Figure S8 to compare the differences in fire [PM$_{2.5}$] between the two inventories at various percentiles. This comparison helps explain why the two datasets exhibited comparable performance for fire episodes, despite a large difference in their mean values: "The probability density distributions of fire-sourced [PM$_{2.5}$] from the two inventories show notable differences (Fig. S8). During 2000-2023, fire [PM$_{2.5}$] from QFED is more than twice that from GFED below the 75$^{th}$ percentile, indicating that QFED predicts significantly higher [PM$_{2.5}$] for low to moderate fire events. However, this difference diminishes above the 90$^{th}$ percentile and becomes particularly constrained at the 99$^{th}$ percentile, where fire-sourced [PM$_{2.5}$] from GFED is 79.29% of that from QFED. It suggests that while both inventories yield comparable

estimates for extreme fire episodes, GFED systematically underestimates emissions from smaller fires. This underestimation persists despite improvements in GFED's representation of small fires through additional implementations (Van Der Werf et al., 2017). Consequently, validations in the U.S. reveal substantial low values with GFED relative to observations (Fig. 4), whereas both inventories perform comparably during high-emission fire episodes (Figs. 3 and S7)." (Lines 244-254)

[Figure]

**Figure S8.** Comparison of daily fire-sourced [$PM_{2.5}$] at different percentiles between simulations with GFED and QFED inventories.

*3) a better quantification and discussion of the uncertainty of their datasets.*

➢ In the revised version, we analyzed of the long-term trend of fire-sourced [$PM_{2.5}$] exceeding the WHO health standard with shadings to quantify the uncertainties from two inventories (Figure 8). We also added following discussion to explain the possible causes of the uncertainties in fire emission inventories: "The two datasets derived from different inventories showed discrepancies in both the long-term mean and trend of fire-sourced [$PM_{2.5}$] (Fig. 5). In general, fire-related [$PM_{2.5}$] is much higher when using the QFED inventory compared to GFED, but the long-term trend is more negative with QFED. As expected, these discrepancies can be attributed to differences in the underlying fire emission inventories (Fig. 6), which stem from variations in their estimation methods, data sources, emission factors, and so on (Kaiser et al., 2012; Larkin et al., 2014; Jin et al., 2023). For example, QFED adjusts emission factors based on aerosol optical depth from MODIS (Petrenko et al., 2012; Li et al., 2022), resulting in significantly higher emissions in some regions compared to GFED. In contrast, GFED relies on burning pixels and changes in surface reflectance identified during satellite overpasses under

relatively cloud-free conditions, which may lead to underestimating burned areas especially for some small fires (Pan et al., 2020)." (Lines 347-357)

In the revised discussion, we explicitly outlined how our study made further progresses compared to Xu et al. (2023):

"We employed a similar approach to Xu et al. (2023) but incorporated new datasets and perspectives. First, we used global observed $PM_{2.5}$ concentrations from 9541 monitoring sites, significantly more than the 5661 stations used in Xu et al. (2023). The expansion of ground-based stations, particularly in fire-prone regions such as Africa and South America, strengthens the foundation for model training and data validation. Second, we applied two different fire emission inventories. Comparisons showed that fire [$PM_{2.5}$] estimates from these inventories were consistent during extreme fire episodes (Figs 3 and S7). However, for low to moderate fire emissions, fire [$PM_{2.5}$] from GFED was much lower than that from QFED (Fig. S8), suggesting that global population exposure to fire-related air pollution may have been underestimated in Xu et al. (2023) due to the application of GFED. Third, we extended the ending simulation year from 2019 to 2023, capturing an additional four years that included unprecedent fire events, such as the 2023 Canadian fires and the 2022 Brazilian fires. These events provide valuable data for assessing population exposure and associated health impacts. Fourth, we found a global decreasing trend in fire [$PM_{2.5}$] during 2000-2023, which contrasts with the increasing trend reported in Xu et al. (2023). This discrepancy may stem from differences in machine learning approaches (random forest vs. XGBoost in this study), pollution definitions (population-weighted vs. non-weighted), and observational datasets. Despite these differences, both studies identified a turning point in 2017, after which global fire [$PM_{2.5}$] began to increase, with the most pronounced rise observed in boreal regions." (Lines 330-346)

*Other comments:*
*1) One recent study found that GFED emissions inventory had a large positive bias in the western US in 2020. This raises potential concerns about the author's methodology of using model-based fire/total ratios to derive their fire-source $PM_{2.5}$ estimates. I think the paper could benefit from a more detailed discussion of this assumption.*
*https://pubs.acs.org/doi/10.1021/acs.est.4c05922*

➢ In the revised paper, we added following discussion to acknowledge the

uncertainties from chemical transport model: "Third, biases in the [PM$_{2.5}$] simulated by the GC model may significantly affect the accuracy of machine learning. Predicting air pollutants involves uncertainties due to variations in meteorological forcing, chemical and physical schemes, initial and boundary conditions, and so on. For example, Qiu et al. (2024) found that the GC model significantly overestimated [PM$_{2.5}$] during extreme wildfire events in 2020 over the western U.S. In contrast, our derived fire [PM$_{2.5}$], using the same GFED inventory, is much lower than the estimates of Childs et al. (2022) for low to median fire events (Fig. 4). These findings suggest that incorporating more validated fire inventories and/or chemical models is necessary to better quantify the uncertainties in derived air pollutant concentrations." (Lines 378-386)

*2) Related to the comment above, I think the paper could benefit from a comparison with the more refined regional fire smoke PM$_{2.5}$ estimates. For example, the authors could compare their two estimates with data from Childs et al (which was recently updated to include 2021-2023) in North America.*

*https://www.stanfordecholab.com/wildfire_smoke*

➢ Thank you for your valuable suggestion. In the revised version, we added Figure 4 and related descriptions to validate the derived fire [PM$_{2.5}$] from both inventories against estimates from Childs et al. (2022): "We further compare the fire-sourced [PM$_{2.5}$] data with the estimates by Childs et al. (2022) in the U.S. (Fig. 4). Our estimates show reasonable performance, with correlation coefficients of 0.68 (0.6) and RMSE of 2.79 (2.71) μg m$^{-3}$ using the GFED (QFED) inventory. However, fire-sourced [PM$_{2.5}$] form GFED is overall lower than that of Childs et al. (2022) by -55.04%." (Lines 239-243)

[Figure]

**Figure 4.** Comparison of fire-sourced PM$_{2.5}$ (μg m$^{-3}$) estimated using (a) GFED and (b) QFED inventories with smoke PM$_{2.5}$ observed by Childs et al. (2022) at 100156 polygons in U.S. during 2016–2019. Validation metrics of N, regression equation, R$^{2}$, RMSE, and NMB are calculated.

*3) The difference between GFED and QFED-based estimates is so large that I think it warrants a more in-depth discussion of the potential reasons. Could the authors draw on prior research that evaluates these emission inventories to discuss the potential reasons?*

*For example: https://doi.org/10.5194/acp-20-969-2020*

➢ In the revised version, we added following discussion to explain the possible causes of the uncertainties in fire emission inventories: "The two datasets derived from different inventories showed discrepancies in both the long-term mean and trend of fire-sourced [$PM_{2.5}$] (Fig. 5). In general, fire-related [$PM_{2.5}$] is much higher when using the QFED inventory compared to GFED, but the long-term trend is more negative with QFED. As expected, these discrepancies can be attributed to differences in the underlying fire emission inventories (Fig. 6), which stem from variations in their estimation methods, data sources, emission factors, and so on (Kaiser et al., 2012; Larkin et al., 2014; Jin et al., 2023). For example, QFED adjusts emission factors based on aerosol optical depth from MODIS (Petrenko et al., 2012; Li et al., 2022), resulting in significantly higher emissions in some regions compared to GFED. In contrast, GFED relies on burning pixels and changes in surface reflectance identified during satellite overpasses under relatively cloud-free conditions, which may lead to underestimating burned areas especially for some small fires (Pan et al., 2020)." (Lines 347-357)

*4) Generating these two datasets (GFED and QFED-based) is an interesting contribution that allows researchers to evaluate the potential uncertainty. However, downstream users often just want to use the best available dataset. What can the authors say in terms of which one they recommend more or less? Also, I think the Xu et al. Nature 2023 paper considered other emissions inventories in their sensitivity analyses, why did the authors decide to focus only on GFED and QFED in this work?*

➢ Xu et al. (2023) compared the uncertainties in fire-sourced [$PM_{2.5}$] from different inventories, but only for the year 2012. In contrast, our study provided datasets from both inventories spanning 2000-2023, enabling us to compare their spatiotemporal variations. Initially, we considered using the FINN and GFAS inventories as well. However, these datasets were slower to update and had incomplete emission data for the timeframe of our study. The GFED and QFED inventories, being more efficient in the data update, were ultimately chosen. Based

on our evaluations, we find that derived fire [$PM_{2.5}$] from both inventories are reasonable (though with some biases). "Therefore, we recommend using the average of fire-sourced [$PM_{2.5}$] from both inventories to indicate the mean state, while using their difference as the range of uncertainties associated with fire-related air pollutants." (Lines 364-366)

**References**

Childs, M. L., Li, J., Wen, J., Heft-Neal, S., Driscoll, A., Wang, S., Gould, C. F., Qiu, M., Burney, J., and Burke, M.: Daily Local-Level Estimates of Ambient Wildfire Smoke $PM_{2.5}$ for the Contiguous US, Environmental Science & Technology, 56, 13607-13621, https://doi.org/10.1021/acs.est.2c02934, 2022.

Jin, L., Permar, W., Selimovic, V., Ketcherside, D., Yokelson, R. J., Hornbrook, R. S., Apel, E. C., Ku, I. T., Collett Jr, J. L., Sullivan, A. P., Jaffe, D. A., Pierce, J. R., Fried, A., Coggon, M. M., Gkatzelis, G. I., Warneke, C., Fischer, E. V., and Hu, L.: Constraining emissions of volatile organic compounds from western US wildfires with WE-CAN and FIREX-AQ airborne observations, Atmos. Chem. Phys., 23, 5969-5991, https://doi.org/10.5194/acp-23-5969-2023, 2023.

Kaiser, J. W., Heil, A., Andreae, M. O., Benedetti, A., Chubarova, N., Jones, L., Morcrette, J. J., Razinger, M., Schultz, M. G., Suttie, M., and van der Werf, G. R.: Biomass burning emissions estimated with a global fire assimilation system based on observed fire radiative power, Biogeosciences, 9, 527-554, https://doi.org/10.5194/bg-9-527-2012, 2012.

Larkin, N. K., Raffuse, S. M., and Strand, T. M.: Wildland fire emissions, carbon, and climate: U.S. emissions inventories, Forest Ecology and Management, 317, 61-69, https://doi.org/10.1016/j.foreco.2013.09.012, 2014.

Li, F., Zhang, X., Kondragunta, S., Lu, X., Csiszar, I., and Schmidt, C. C.: Hourly biomass burning emissions product from blended geostationary and polar-orbiting satellites for air quality forecasting applications, Remote Sensing of Environment, 281, 113237, https://doi.org/10.1016/j.rse.2022.113237, 2022.

Pan, X., Ichoku, C., Chin, M., Bian, H., Darmenov, A., Colarco, P., Ellison, L., Kucsera, T., da Silva, A., Wang, J., Oda, T., and Cui, G.: Six global biomass burning emission datasets: intercomparison and application in one global aerosol model, Atmos. Chem. Phys., 20, 969-994, https://doi.org/10.5194/acp-20-969-2020, 2020.

Petrenko, M., Kahn, R., Chin, M., Soja, A., Kucsera, T., and Harshvardhan: The use of satellite-measured aerosol optical depth to constrain biomass burning emissions source strength in the global model GOCART, Journal of Geophysical Research: Atmospheres, 117, https://doi.org/10.1029/2012JD017870, 2012.

Qiu, M., Kelp, M., Heft-Neal, S., Jin, X., Gould, C. F., Tong, D. Q., and Burke, M.: Evaluating Chemical Transport and Machine Learning Models for Wildfire Smoke $PM_{2.5}$: Implications for Assessment of Health Impacts, Environmental Science & Technology, 58, 22880-22893, https://doi.org/10.1021/acs.est.4c05922, 2024.

van der Werf, G. R., Randerson, J. T., Giglio, L., van Leeuwen, T. T., Chen, Y., Rogers, B. M., Mu, M., van Marle, M. J. E., Morton, D. C., Collatz, G. J., Yokelson, R. J.,

and Kasibhatla, P. S.: Global fire emissions estimates during 1997–2016, Earth Syst. Sci. Data, 9, 697-720, https://doi.org/10.5194/essd-9-697-2017, 2017.

Xu, R., Ye, T., Yue, X., Yang, Z., Yu, W., Zhang, Y., Bell, M. L., Morawska, L., Yu, P., Zhang, Y., Wu, Y., Liu, Y., Johnston, F., Lei, Y., Abramson, M. J., Guo, Y., and Li, S.: Global population exposure to landscape fire air pollution from 2000 to 2019, Nature, 621, 521-529, https://doi.org/10.1038/s41586-023-06398-6, 2023.

---

## Author Comment (AC3)

We are grateful to the editor and referee for their time and energy in providing helpful comments and guidance that have improved the manuscript. In this document, we describe how we have addressed the reviewer's comments. Referee comments are shown in black italics and author responses are shown in blue regular text.

*Reviewer #3:*

*The study has constructed a global dataset of fire-sourced $PM_{2.5}$ concentrations at a spatial resolution of 0.25 degree and daily scale covering the period of 2000-2023. The dataset is developed using global model simulations with two fire emission inventories, and further with bias-corrected using a machine learning algorithm applied to predict global surface $PM_{2.5}$ measurements. Differences between the fire-sourced $PM_{2.5}$ concentrations derived from the two fire emission inventories (GFED vs. QFED) are further analyzed.*

*Overall, I think the study is well conducted and the high-resolution fire-sourced $PM_{2.5}$ dataset is valuable for the community to further explore the impacts of fires on the environment, such as its use in the recent study (Xu et al., Nature 2023). Publishing the dataset (with reasonable extension relative to the previous study) on ESSD appears to fit the journal's scope.*

*Here I have several comments on the quality of the dataset that hope the authors can further address and refine.*

➢ Thank you for your positive evaluations.

*Comments*

*1) The differences between GFED vs. QFED derived fire $PM_{2.5}$ need to be better quantified. It seems that compared with the previous dataset of Xu et al. (Nature 2023), the datasets presented in this study apply two different fire emission inventories. Why were the two fire emission inventories selected? Did they represent the current fire emission uncertainty ranges? Section 3.3 discussed their differences but mainly focused on the mean values. How about the episodic fire events? Some comparisons based on the daily scale would be valuable.*

➢ Thank you for your valuable suggestions. Xu et al. (2023) compared the uncertainties in fire-sourced [$PM_{2.5}$] from different inventories, but only for the year 2012. In contrast, our study provided datasets from both inventories spanning

2000-2023, enabling us to compare their spatiotemporal variations. Initially, we considered using the FINN and GFAS inventories as well. However, these datasets were slower to update and had incomplete emission data for the timeframe of our study. The GFED and QFED inventories, being more efficient in the data update, were ultimately chosen.

The inclusion of two inventories is one of the major contributions of our study to the community. In the revised version, we added Figure 4 to validate the derived fire [$PM_{2.5}$] from both inventories against estimates from Childs et al. (2022), and Figure S8 to compare the differences between them at various percentiles: "The probability density distributions of fire-sourced [$PM_{2.5}$] from the two inventories show notable differences (Fig. S8). During 2000-2023, fire [$PM_{2.5}$] from QFED is more than twice that from GFED below the 75th percentile, indicating that QFED predicts significantly higher [$PM_{2.5}$] for low to moderate fire events. However, this difference diminishes above the 90th percentile and becomes particularly constrained at the 99th percentile, where fire-sourced [$PM_{2.5}$] from GFED is 79.29% of that from QFED. It suggests that while both inventories yield comparable estimates for extreme fire episodes, GFED systematically underestimates emissions from smaller fires. This underestimation persists despite improvements in GFED's representation of small fires through additional implementations (Van Der Werf et al., 2017). Consequently, validations in the U.S. reveal substantial low values with GFED relative to observations (Fig. 4), whereas both inventories perform comparably during high-emission fire episodes (Figs. 3 and S7)." (Lines 244-254)

[Figure]

**Figure 4.** Comparison of fire-sourced $PM_{2.5}$ (µg m$^{-3}$) estimated using (a) GFED and (b) QFED inventories with smoke $PM_{2.5}$ observed by Childs et al. (2022) at 100156 polygons in U.S. during 2016–2019. Validation metrics of N, regression equation, $R^2$, RMSE, and NMB are calculated.

[Figure]

**Figure S8.** Comparison of daily fire-sourced [PM$_{2.5}$] at different percentiles between simulations with GFED and QFED inventories.

In the revised paper, we expanded our discussion on the causes of these uncertainties and offered recommendations on how to best use these datasets: "The two datasets derived from different inventories showed discrepancies in both the long-term mean and trend of fire-sourced [PM$_{2.5}$] (Fig. 5). In general, fire-related [PM$_{2.5}$] is much higher when using the QFED inventory compared to GFED, but the long-term trend is more negative with QFED. As expected, these discrepancies can be attributed to differences in the underlying fire emission inventories (Fig. 6), which stem from variations in their estimation methods, data sources, emission factors, and so on (Kaiser et al., 2012; Larkin et al., 2014; Jin et al., 2023). For example, QFED adjusts emission factors based on aerosol optical depth from MODIS (Petrenko et al., 2012; Li et al., 2022), resulting in significantly higher emissions in some regions compared to GFED. In contrast, GFED relies on burning pixels and changes in surface reflectance identified during satellite overpasses under relatively cloud-free conditions, which may lead to underestimating burned areas especially for some small fires (Pan et al., 2020). Further validations showed that all-source [PM$_{2.5}$] using GFED yielded an R value of 0.58±0.29 and an NMB of 10.68±24.96% averaged for the 12 fire episodes (Fig. 3). Slightly improved statistical metrics were achieved using QFED, with an R value of 0.63±0.26 and an NMB of 6.56±27.61% for the same events (Fig. S7). However, these differences are too minor to conclusively determine which dataset provides a better estimate of fire-sourced [PM$_{2.5}$]. Fire-sourced [PM$_{2.5}$] is generally lower in the GFED dataset compared to QFED; exceptions exist, such as the 2023 Canadian fires, in which fire-sourced [PM$_{2.5}$] from GFED (Fig. 7) was significantly higher than that

from QFED (Fig. S9). Therefore, we recommend using the average of fire-sourced [PM$_{2.5}$] from both inventories to indicate the mean state, while using their difference as the range of uncertainties associated with fire-related air pollutants." (Lines 347-366)

As for the episodic fire events, we analyzed of the long-term trend of fire-sourced [PM$_{2.5}$] exceeding the WHO health standard with shadings to quantify the uncertainties from two inventories (Figure 8). Along with other figures (e.g., Figs 3 vs. S7, Figs 7 vs. S9), our study provided a thorough comparison and validation of fire [PM$_{2.5}$] during extreme events derived from two inventories.

 "Extreme fire episodes pose significant threats to public health. The percentage of days and land grids with fire-sourced [PM$_{2.5}$] exceeding the World Health Organization's air quality standard of 15 μg m$^3$ showed a global decreasing trend of -0.03% yr$^{-1}$ (Fig. 8a). Regionally, an increase of 0.04% yr$^{-1}$ was found in North America, driven by the 2023 Canadian fire episode, though this change was not statistically significant. In other regions, the exposure risk to high levels of fire PM$_{2.5}$ declines, with the most notable declines of -0.22% yr$^{-1}$ in South America and -0.13% yr$^{-1}$ in Africa. While extreme fire [PM$_{2.5}$] in general decreased, a turning point occurred in 2017, with more pronounced fire events thereafter." (Lines 289-296)

[Figure]

**Figure 8.** Annual percentage of days and land grids with fire-sourced [$PM_{2.5}$] exceeding 15 μg m³ in (a) Global, (b) Asia, (c) North America, (d) Africa, (e) South America and (f) Oceania for 2000-2023. The average estimates from GFED and QFED are shown as bold lines, with shadings indicating their range. Regional trends are displayed on the top right of each panel, with an asterisk denoting significant ($p < 0.05$) changes.

*2) As shown in Figure 4 and Figure S3, the model simulated all-source $PM_{2.5}$ concentrations were significantly biased high over many regions. How would these model biases affect the fire-sourced $PM_{2.5}$ estimates? According to Equ (3) in Section 2.4, if the biases were from the nofire model $PM_{2.5}$, then the resulting fire-sourced $PM_{2.5}$ would be underestimated. The impacts of the model biases shall be better discussed.*

➢ In the revised paper, we added following discussion to acknowledge the uncertainties from chemical transport model: "Third, biases in the [$PM_{2.5}$] simulated by the GC model may significantly affect the accuracy of machine learning. Predicting air pollutants involves uncertainties due to variations in meteorological forcing, chemical and physical schemes, initial and boundary conditions, and so on. For example, Qiu et al. (2024) found that the GC model significantly overestimated [$PM_{2.5}$] during extreme wildfire events in 2020 over the western U.S. In contrast, our derived fire [$PM_{2.5}$], using the same GFED inventory, is much lower than the estimates of Childs et al. (2022) for low to median fire events (Fig. 4). These findings suggest that incorporating more validated fire inventories and/or chemical models is necessary to better quantify the uncertainties in derived air pollutant concentrations." (Lines 378-386)

**References**

Childs, M. L., Li, J., Wen, J., Heft-Neal, S., Driscoll, A., Wang, S., Gould, C. F., Qiu, M., Burney, J., and Burke, M.: Daily Local-Level Estimates of Ambient Wildfire Smoke $PM_{2.5}$ for the Contiguous US, Environmental Science & Technology, 56, 13607-13621, https://doi.org/10.1021/acs.est.2c02934, 2022.

Jin, L., Permar, W., Selimovic, V., Ketcherside, D., Yokelson, R. J., Hornbrook, R. S., Apel, E. C., Ku, I. T., Collett Jr, J. L., Sullivan, A. P., Jaffe, D. A., Pierce, J. R., Fried, A., Coggon, M. M., Gkatzelis, G. I., Warneke, C., Fischer, E. V., and Hu, L.: Constraining emissions of volatile organic compounds from western US wildfires with WE-CAN and FIREX-AQ airborne observations, Atmos. Chem. Phys., 23, 5969-5991, https://doi.org/10.5194/acp-23-5969-2023, 2023.

Kaiser, J. W., Heil, A., Andreae, M. O., Benedetti, A., Chubarova, N., Jones, L., Morcrette, J. J., Razinger, M., Schultz, M. G., Suttie, M., and van der Werf, G. R.: Biomass burning emissions estimated with a global fire assimilation system based on observed fire radiative power, Biogeosciences, 9, 527-554, https://doi.org/10.5194/bg-9-527-2012, 2012.

Larkin, N. K., Raffuse, S. M., and Strand, T. M.: Wildland fire emissions, carbon, and climate: U.S. emissions inventories, Forest Ecology and Management, 317, 61-69, https://doi.org/10.1016/j.foreco.2013.09.012, 2014.

Li, F., Zhang, X., Kondragunta, S., Lu, X., Csiszar, I., and Schmidt, C. C.: Hourly biomass burning emissions product from blended geostationary and polar-orbiting satellites for air quality forecasting applications, Remote Sensing of Environment, 281, 113237, https://doi.org/10.1016/j.rse.2022.113237, 2022.

Pan, X., Ichoku, C., Chin, M., Bian, H., Darmenov, A., Colarco, P., Ellison, L., Kucsera, T., da Silva, A., Wang, J., Oda, T., and Cui, G.: Six global biomass burning emission datasets: intercomparison and application in one global aerosol model, Atmos. Chem. Phys., 20, 969-994, https://doi.org/10.5194/acp-20-969-2020, 2020.

Petrenko, M., Kahn, R., Chin, M., Soja, A., Kucsera, T., and Harshvardhan: The use of satellite-measured aerosol optical depth to constrain biomass burning emissions source strength in the global model GOCART, Journal of Geophysical Research: Atmospheres, 117, https://doi.org/10.1029/2012JD017870, 2012.

Qiu, M., Kelp, M., Heft-Neal, S., Jin, X., Gould, C. F., Tong, D. Q., and Burke, M.: Evaluating Chemical Transport and Machine Learning Models for Wildfire Smoke $PM_{2.5}$: Implications for Assessment of Health Impacts, Environmental Science & Technology, 58, 22880-22893, https://doi.org/10.1021/acs.est.4c05922, 2024.

van der Werf, G. R., Randerson, J. T., Giglio, L., van Leeuwen, T. T., Chen, Y., Rogers, B. M., Mu, M., van Marle, M. J. E., Morton, D. C., Collatz, G. J., Yokelson, R. J., and Kasibhatla, P. S.: Global fire emissions estimates during 1997–2016, Earth Syst. Sci. Data, 9, 697-720, https://doi.org/10.5194/essd-9-697-2017, 2017.

---

## Author Response (AR2)

We sincerely thank the editor and reviewers once again for their great efforts and constructive comments. In this document, we outline our responses to the second-round comments. Reviewer comments are shown in black italics, and our responses are provided in blue regular text. A manuscript with tracking changes is attached at the end.

*Reviewer #1:*

*1) Please discuss the long-term trends in fire-PM$_{2.5}$ in the context of long-term trends in the used emission inventory (GFED and QFED).*

➢ In the revised version, we added discussion about the long-term trends in fire-PM$_{2.5}$ with GFED and QFED inventories: "These differences in fire-sourced [PM$_{2.5}$] are mainly due to the discrepancies in fire inventories. In global fire-prone regions, organic carbon (OC) emissions from fires are 51.08-65.18% lower in the GFED inventory compared to the QFED inventory (Fig. 6a). Consequently, the global average fire-sourced [PM$_{2.5}$] is estimated at 2.04 μg m$^{-3}$ with GFED, nearly half of the 3.96 μg m$^{-3}$ estimated with QFED (Table 2). Moreover, fire emission trends in QFED tend to be more negative or less positive than in GFED (Fig. 6b), leading to stronger negative trends in fire-sourced [PM$_{2.5}$] derived from QFED (Fig. 6d). For both inventories, simulated fire [PM$_{2.5}$] trends are more negative than the corresponding emission trends, likely due to climatic or chemical conditions that enhance pollutant removal. For example, in North America, increased atmospheric oxidant levels (e.g., increased OH and O$_3$) and changes in boundary layer height over the past two decades may have offset rising fire emissions by accelerating aerosol aging and modifying vertical mixing (Heilman et al., 2014;Zhou et al., 2019). In Siberia, the positive trend in GFED emissions is not fully reflected in fire-sourced [PM$_{2.5}$], likely due to concurrent increases in rainfall and deposition efficiency that enhance particulate scavenging (Konovalov et al., 2024)." (Lines 269-282)

*2) Figure 4 is interesting to show higher uncertainty of the estimates at low concentrations. Could you bin the pairs based on the Child et al. (2019) data in different concentration levels and provide evaluation statistics for different bins? I anticipate this to be a more valuable quantitative constraints of the uncertainty of the estimates at*

*different levels of fire-PM$_{2.5}$.*

➢ In the revised version, we added Figure S9 to quantify the discrepancy between our estimates and Childs et al. at different bins: "Consequently, validations in the U.S. reveal substantial low values with GFED relative to previous estimates (Fig. 4), a bias that is alleviated in QFED for small to moderate fires (Fig. S9). Although both inventories perform comparably during high-emission events (Figs. 3 and S7), their estimates remain much lower than those of Childs et al. (2022) at the highest levels of fire-sourced [PM$_{2.5}$] (Fig. S9). " (Lines 251-255)

[Figure]

**Figure S9.** Boxplot of estimated GFED and QFED fire PM$_{2.5}$ v.s. Childs et al. (2022) estimated smoke PM$_{2.5}$ under various levels.

*3) Figure 5b,d,f. My original doubt was: why are these slashes spaced so distant and regularly? Could you plot the map of p values to check if you are actually labeling the correct locations with significant trends?*

➢ In the revised version, we replotted Figure 5 with denser slashes. We also added Figure S10 to indicate the *p* values of trends.

[Figure]

**Figure 5.** Long-term (a) mean and (b) trend of fire [PM$_{2.5}$] (µg m$^{-3}$) derived using the GFED inventory for 2000-2023. The box regions in (a) indicate areas used for comparing differences between two inventories. Panels (c) and (d) display the same information as (a) and (b), but for fire [PM$_{2.5}$] from QFED inventory. The differences in fire [PM$_{2.5}$] (Δ[PM$_{2.5}$]) between the two inventories are presented for the long-term (e) mean and (f) trend during 2000-2023. Green slashes indicate areas with significant ($p < 0.05$) changes. The $p$ values of these trend are shown in Fig. S10.

[Figure]

**Figure S10.** The *p* values of long-term (a) trends in fire [$PM_{2.5}$] derived using the GFED inventory for 2000-2023. Panels (b) and (c) display the same information as (a), but for fire [$PM_{2.5}$] from QFED inventory and differences between QFED and GFED inventories.

**References**

Basart, S., Pérez, C., Cuevas, E., Baldasano, J. M., and Gobbi, G. P.: Aerosol characterization in Northern Africa, Northeastern Atlantic, Mediterranean Basin and Middle East from direct-sun AERONET observations, Atmos. Chem. Phys., 9, 8265-8282, https://doi.org/10.5194/acp-9-8265-2009, 2009.

Heilman, W. E., Liu, Y., Urbanski, S., Kovalev, V., and Mickler, R.: Wildland fire emissions, carbon, and climate: Plume rise, atmospheric transport, and chemistry processes, Forest Ecology and Management, 317, 70-79, https://doi.org/10.1016/j.foreco.2013.02.001, 2014.

Konovalov, I. B., Golovushkin, N. A., and Beekmann, M.: Wildfire-smoke-precipitation interactions in Siberia: Insights from a regional model study, Science of The Total Environment, 951, 175518,

https://doi.org/10.1016/j.scitotenv.2024.175518, 2024.

Zhou, S., Collier, S., Jaffe, D. A., and Zhang, Q.: Free tropospheric aerosols at the Mt. Bachelor Observatory: more oxidized and higher sulfate content compared to boundary layer aerosols, Atmos. Chem. Phys., 19, 1571-1585, https://doi.org/10.5194/acp-19-1571-2019, 2019.